# Functional host-specific adaptation of the intestinal microbiome in hominids

M. C. Rühlemann [1,2] ✉, C. Bang[1], J. F. Gogarten[3,4,5,6], B. M. Hermes[7,8], M. Groussin [1], S. Waschina [9], M. Poyet[8], M. Ulrich[4,5], C. Akoua-Koffi[10], T. Deschner[11], J. J. Muyembe-Tamfum[12], M. M. Robbins[13], M. Surbeck [14,15], R. M. Wittig[16,17], K. Zuberbühler[18,19], J. F. Baines[7,8], F. H. Leendertz[4,5] & A. Franke [1] ✉

Fine-scale knowledge of the changes in composition and function of the human gut microbiome compared that of our closest relatives is critical for understanding the evolutionary processes underlying its developmental trajectory. To infer taxonomic and functional changes in the gut microbiome across hominids at different timescales, we perform high-resolution metagenomic-based analyzes of the fecal microbiome from over two hundred samples including diverse human populations, as well as wild-living chimpanzees, bonobos, and gorillas. We find human-associated taxa depleted within non-human apes and patterns of host-specific gut microbiota, suggesting the widespread acquisition of novel microbial clades along the evolutionary divergence of hosts. In contrast, we reveal multiple lines of evidence for a pervasive loss of diversity in human populations in correlation with a high Human Development Index, including evolutionarily conserved clades. Similarly, patterns of co-phylogeny between microbes and hosts are found to be disrupted in humans. Together with identifying individual microbial taxa and functional adaptations that correlate to host phylogeny, these findings offer insights into specific candidates playing a role in the diverging trajectories of the gut microbiome of hominids. We find that repeated horizontal gene transfer and gene loss, as well as the adaptation to transient microaerobic conditions appear to have played a role in the evolution of the human gut microbiome.

Human gut microbiome research has demonstrated that numerous factors, including diet, environment, and lifestyle influence the structure of the human gut microbiota, which in turn have profound impacts on human health and disease[1–3]. To date, the majority of these studies were conducted on sample collections from high-income countries, however growing efforts to include humans from diverse global populations are underway, thereby providing an additional angle to investigate and evaluate shared and specific microbiome properties across human populations[3–6].

These efforts provided an opportunity to discover signatures of host geography and lifestyle that go beyond conventional differences in diversity parameters in the gut community. For instance, they revealed elevated rates of horizontal gene transfer (HGT) that correlate with the Human Development Index (HDI, a statistical composite index of indicators encompassing life expectancy, education, and income[7];) of a population, suggesting that gut microbiota constantly acquire new functions in conjunction with host lifestyle changes[8]. However, the mechanisms that link changes in

gut microbial structure with host behavior and ecology remain largely unexplored.

Additional critical insight into understanding patterns of diversity and composition among human gut communities can be obtained from comparative surveys of the hominid gut microbiota. Humans, chimpanzees, bonobos, and gorillas show increasingly divergent gut microbiota, with more distantly related species exhibiting more divergent community composition (phylosymbiosis[9];). At the same time, the phylogeny of some of their individual microbial lineages parallels their own phylogeny (codivergence[10,11];). Both patterns of phylosymbiosis and codivergence are suggestive of long-term effects of hominid evolution on their communities of symbionts[12–14]. Notably, results from comparative marker-gene analyzes suggest co-diversifying members of the hominid gut microbial communities (both prokaryotic and phage) are lost and replaced with human lineages when animals leave their natural environments and are moved into captivity[15,16]. However, poor taxonomic resolution and a lack of functional characterization precludes a deeper understanding of processes driving these changes.

Functional analyzes from shotgun metagenomic data revealed a conserved phylogenetic signal across wild non-human primates (NHP), despite dietary changes over an individual's lifespan and between species, suggesting that the evolution of gut microbiota within wild NHP is partially constrained by host genetics and physiology[17]. A comparative meta-analysis of microbial functions in NHP and diverse human populations observed a comparable loss of biodiversity in captive NHP and human populations from regions with higher HDI[18], supporting previous findings[19]. However, overall, the functional characterization of NHP microbiota in previous studies has been limited to only selected microbial taxa without addressing broader-scale functional changes and specific alterations, especially in African great apes and humans. As such, robust comparative functional analyzes are still needed for a comprehensive understanding of how gut microbiota have evolved with hominids and shaped the current structure and functional capabilities (and deficits) of the human gut microbiome.

To better elucidate host-microbiome interactions in the hominid gut in an evolutionary context, we present a large-scale comparative study of wild non-human apes (NHA) and humans from geographically distinct populations spanning two continents. Functional shotgun metagenomic sequencing was performed on feces samples of wild-living great apes from six African countries, including two gorilla species, three chimpanzee subspecies, and bonobos, and combined with published data from gorillas and chimpanzees from the Republic of Congo[20]. Additionally, we sequenced human fecal samples from two African populations[21] from rural villages of the Taï region in Côte d'Ivoire (HDI$_{2021}$ = 0.550[7];) and the Bandundu region near Salonga National Park, Democratic Republic of the Congo (HDI$_{2021}$ = 0.479), along with samples from Germany (HDI$_{2021}$ = 0.942), and included a published dataset from Denmark (HDI$_{2021}$ = 0.948[22];) to incorporate varying degrees of HDI. Using this extensive data resource, we created a comprehensive catalog of high-quality prokaryotic genomes assembled from metagenomic data, which we annotated on a taxonomic and functional level. We subsequently explored patterns of diversity and host-specificity for both taxonomic groups and functions, which reveals intriguing patterns associated with human gut microbial communities, convergent functional adaptations across lineages, and the potential mechanisms driving these patterns.

## Results
### An expanded catalog of microbial genomes from the hominid gut

Using 224 shotgun metagenomic datasets (Suppl. Data 1) from fecal samples of humans (Côte d'Ivoire (CIV), $n$ = 12, Dem. Rep. of the Congo (DRC), $n$ = 12, Denmark (DK), $n$ = 24, and Germany (GER), $n$ = 24) and non-human apes, including two gorilla subspecies (*Gorilla gorilla gorilla*, Gabon (GAB), $n$ = 8; *Gorilla beringei beringei*, Uganda (UGA), $n$ = 11, and Republic of Congo (CG), $n$ = 28), three chimpanzee subspecies (*Pan troglodytes verus*, CIV, $n$ = 55; *P.t. troglodytes*, GAB, $n$ = 11, and CG, $n$ = 18; *P.t. schweinfurthii*, UGA, $n$ = 12), and bonobos (*Pan paniscus*, DRC, $n$ = 12), we reconstructed a total of 7700 metagenome-assembled genomes (MAGs) ensuring maximum completeness and low contamination using multiple binning algorithms and dedicated curation and scoring tools ([23]; see Methods). The most MAGs (quality score >50%) were reconstructed for the most sampled subgroup of great ape, *P.t. verus* ($n$ = 2182), while the average number of reconstructed MAGs per sample was the highest for *P.t. schweinfurthii* (mean=74.5). Library size / number of sequencing reads was highly correlated with total assembly size ($\rho_{Spearman}$ = 0.644), which in turn was directly correlated with the number of bins recovered for a sample ($\rho_{Spearman}$ = 0.966).

To ensure a comprehensive reference for the analysis, the collection of MAGs was combined with two large collection of microbial reference species reconstructed from human fecal metagenomes (UHGGv2, $n$ = 4744 isolates and MAGs[24];), and non-human primate fecal metagenomes ($n$ = 1295 MAGs[18];), resulting in a total of $n$ = 13,739 genome sequences; MAGs were subsequently clustered into 5777 species-level genome bins (SGBs; 95% ANI) using stringent criteria (Suppl. Fig. S1a and S1b, Supplementary Data 2). Of these, 1074 SGBs were not previously covered by either of the two large reference sets, mostly originating from NHA samples ($n$ = 956, 89.0%; Suppl. Fig. S1c). The highest-quality genome sequence in each SGB was chosen as its representative. Overall quality of SGB representative genomes was high (median quality score = 94.1%; Suppl. Fig. S1d). SGB representatives were used as comprehensive reference for the estimation of per-sample abundances (Methods, Suppl. Fig. S1e, Suppl. Data 3). In total 3287 SGBs, encompassing 21 bacterial and two archaeal phyla (Suppl. Fig. S1b), were found present in the dataset. This curated catalog of SGBs from gorillas, bonobos, and chimpanzees increases the number of microbial species genomes previously reconstructed from feces by more than ten-fold, and increases mapping success of fecal metagenomes from NHAs to reference genomes by two- to three-fold (Suppl. Fig. S1e[18]). As expected, only minor proportions of SGBs from human samples were not previously covered by the included large reference collections, with 5.8%, 2.7 %, 1.8 and 1.7% of SGBs found in samples from CIV, DRC, DK, and GER, respectively, falling into this group (Suppl. Fig. S1e). For both NHAs and humans, the highest percentages of previously uncovered diversity were observed within the phyla Bacteroidota and Spirochaetota, and, to a lesser degree, within Firmicutes and Firmicutes A for NHAs only (Suppl. Fig. S1f). Generally, recovered clusters were highly host specific. While the 7700 MAGs spanned 1787 of the final SGBs, only for 48 of these SGBs MAGs were reconstructed from samples of more than a single host genus.

Within sample diversity varied considerably between host (sub-) species and with sequencing depth (Suppl. Fig. S1g). The used SGB collection covers large proportions of the diversity found within the human gut (Suppl. Fig. S1e) and Faith's phylogenetic diversity (PD) incorporates SGB relatedness in the diversity calculation, which enables a better estimate of total diversity of a community then simpler richness estimates from taxonomic group abundances. Phylogenetic diversity (PD) at an even mapping depth of 1 million reads per sample showed significantly lower diversity in humans compared to all African great ape hosts ($P_{Wilcoxon}$ = 1.2 × 10$^{-13}$; Fig. 1a). Comparing individual human populations to African great ape hosts revealed that humans from GER and DRC showed lower means than all NHA group (all $P_{Wilcoxon}$ < 0.05), while humans from CIV and DK exhibited high variance and were found to have significant lower diversity than all NHA hosts ($P_{Wilcoxon}$ < 0.05), except for *G. g. gorilla* and *P.t. troglodytes* ($P_{Wilcoxon}$ > 0.05). These two great ape taxa were found to have the lowest library sizes, low numbers of recovered genomes and lowest

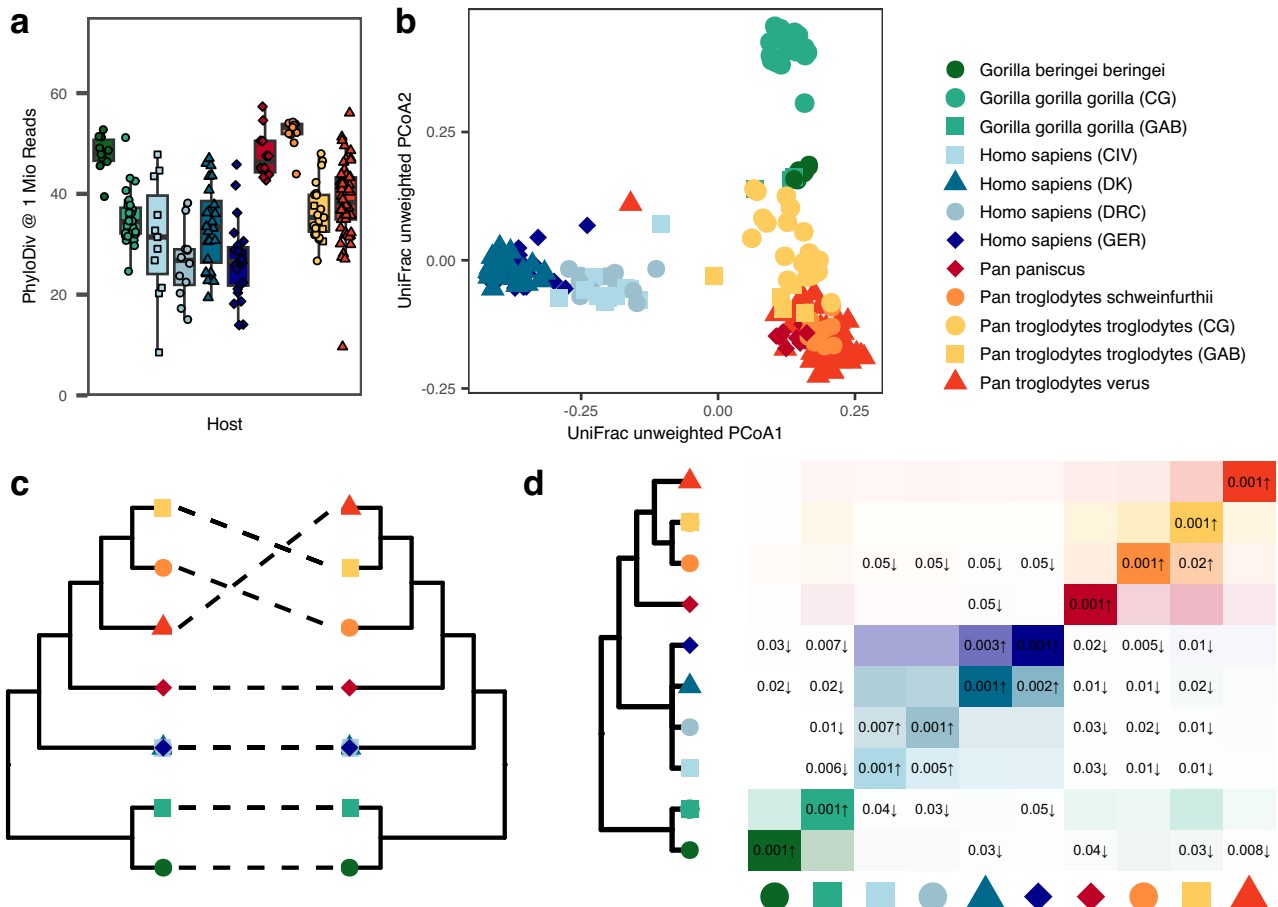

**Fig. 1 | Community-level specificities of human and NHA fecal microbiomes.**
**a** Phylogenetic Diversity across host groups at a sampling depth of 1 Mio. mapped reads per sample. **b** Ordination of unweighted UniFrac distances of all samples, colored by host subgroups. **c** Tanglegram of host (left) and microbiome (right) trees, the latter based on unweighted UniFrac distances. **d** SGB sharing coefficients between host group. Rows represent reference host groups; columns represent the groups with which they share overlap. Numbers in the tiles are *P*-values from the analysis for enrichment (↑) and depletion (↓) in the reference group based on random 1000 permuations (unadjusted, one-sided). All analyzes are based on $n = 211$ independent samples ($n_{Human} = 71$, $n_{Chimpanzee} = 91$, $n_{Bonobo} = 12$, $n_{Gorilla} = 40$). Boxplots show the following elements: center line: median, box limits: upper and lower quartile; whiskers: $1.5 \times$ interquartile ranges.

mapping efficiency. Taken together, this suggests that phylogenetic diversity in these hosts may be biased by lower representation in the reference database and that the reduced diversity could be an artefact of this. Consequently, samples with less than 1 million mapped reads were removed from further analyzes. This resulted in the removal of samples from the analysis for the groups *G.b. beringei* ($n = 1$), *G.g. gorilla* ($n_{GAB} = 5$, $n_{CG} = 1$), *P.t. troglodytes* ($n_{GAB} = 4$), *P.t. schweinfurthii* ($n = 1$) and humans from CIV ($n = 1$). Of note, the lowest and highest mean values for within-sample phylogenetic diversity were found for the human subgroups from Germany and Denmark, respectively, the latter exhibiting the only significant differences between human sub-sets ($P_{Wilcoxon} < 0.05$ vs. DRC and GER), contradicting previous reports of lower alpha diversity generally found in high HDI countries[3,4]. Additionally, the considerable differences in PD observed between CIV and DRC highlight the diversity found between human populations and the need to better characterize human gut microbiome diversity. Therefore, while total diversity of some host groups was likely not exhaustively sampled, especially considering lower abundant microbial clades, the presented reference collection of high-quality meta-genome reconstructed genomes likely represents the current best resource for an in-depth taxonomic and functional assessment of hominid fecal microbiomes and highlights that some human populations, including those sampled in this analysis, have lost considerable microbial diversity in their guts.

## Phylosymbiosis in hominids is strongly supported by community structure

Phylosymbiosis, a pattern in which microbial community divergence parallels that of the hosts, can be a sign of community-level co-evolution of host and microbiota, indicative of host-microbe relationships maintained over evolutionary timescales[9]. To investigate such patterns, we used six different measures of beta diversity. Four were based on phylogenetic or taxonomic distance metrics (weighted and unweighted UniFrac, as well as genus level Aitchison and Jaccard distance), and two additional metrics considered the functional capacities of the community (KEGG ortholog (KO) abundance and presence/absence patterns; Suppl. Fig. 2). Strong signals for phylosymbiosis were found for Jaccard, and unweighted UniFrac distances ($P < 0.001$; $Q_{Bonferroni} < 0.01$), as well as a less pronounced signal for Aitchison distance and the abundance of functional groups ($P < 0.01$, $Q_{Bonferroni} = 0.0534$ and $Q_{Bonferroni} = 0.0585$, respectively; Fig. 1b and c, Suppl. Fig. 2, Suppl. Data 4), but not for the distances based on presence and absence of KOs ($P = 0.055$, $Q_{Bonferroni} = 0.33$) and the weighted UniFrac distance ($P = 1$). These results suggest phylosymbiotic divergence in the general microbial structure of hominid microbiota (Jaccard and unweighted UniFrac are based on the presence/absence of microbial clades) which in part is paralleled by shifted abundances of distinct taxonomic and functional groups, generally in line with previous observations in other host systems[9]. This reduced

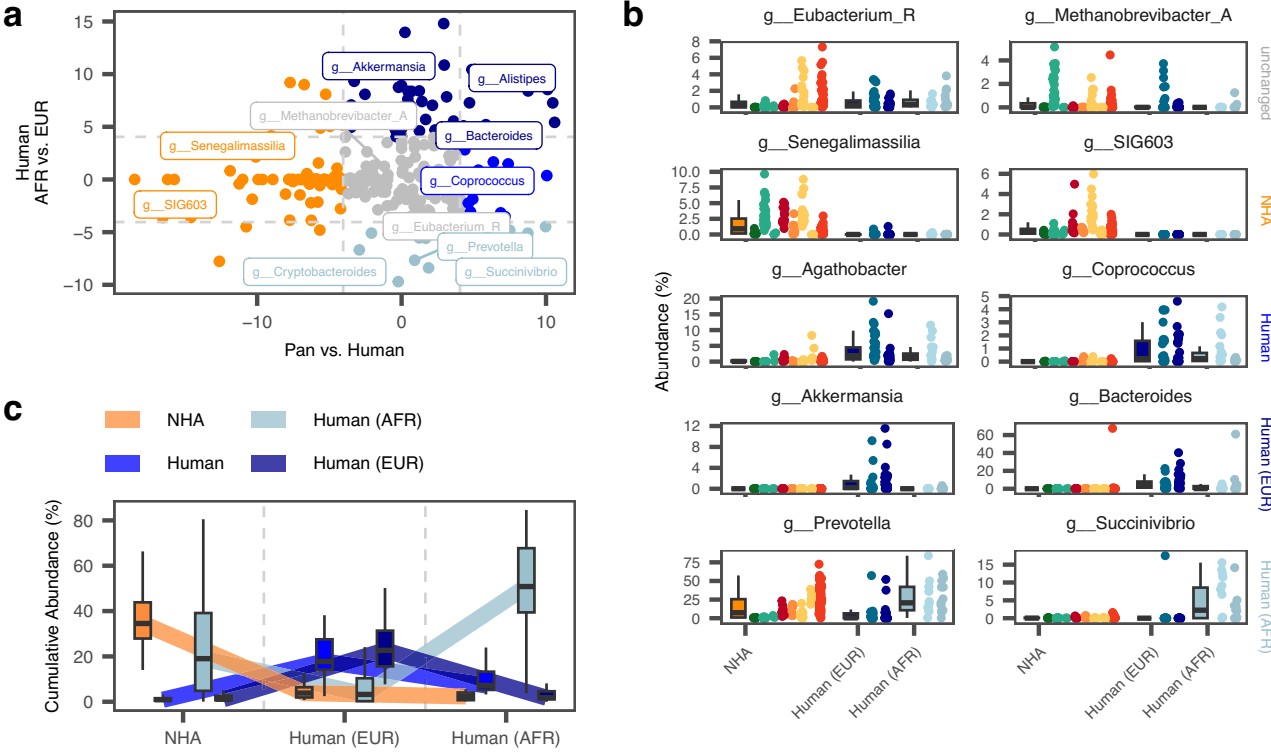

**Fig. 2 | Taxonomic differences the microbiota of humans and NHAs. a** Effect sizes (t-values from univariate linear regression) of abundance differences of genera in fecal samples of NHAs (*n* = 143) and humans (x-axis) and humans within African (*n* = 23) and European (*n* = 48) populations separated as well (y-axis). Points are colored according to the association groups with European (dark blue) and African human populations (light blue), or to indicate enrichment in NHAs (orange) or humans (bright blue). Taxonomic groups not found to be associated with any of the groups (all Q > 0.05, two-sided) are shown in grey. Horizontal and vertical lines depict the t-value threshold ( | t-value | > 4.04) for statistical significance after Bonferroni-correction. **b** Per-sample and host group abundances of selected genera

found with unchanged abundances across all groups (top), increased abundance in NHAs (*n* = 143) or humans (*n* = 71; rows 2 and 3, respectively), or in humans from Africa (*n* = 23) or Europe (*n* = 48; rows 4 and 5). Points are colored according to host genus: Gorilla = greens, Pan = reds, oranges, and yellows, and human = blues. **c** Cumulative abundance trajectories of taxa associated with human communities and NHAs. Shown are the per-sample cumulative abundances within each host group, grouped based on a taxon's association with either NHAs (*n* = 143), all humans(*n* = 71), or one of the human population subgroups. All boxplots show the following elements: center line: median, box limits: upper and lower quartile; whiskers: 1.5 × interquartile ranges.

signal found for taxon-level diversity might be the result of fluctuating clade abundances and functions in response to changing environmental factors, including diet, indicating a high functional plasticity of the hominid gut microbiome in response to immediate influences, while structural changes through acquisition and loss of microbial clades might rather result from longer-term adaptations.

SGBs shared between host groups were analyzed in a permutation-based framework accounting for differences in sequencing depth and group sizes (see Methods for details). We found that especially human-associated SGBs were strongly depleted across multiple NHA groups (Fig. 1d), while such signal found in the opposite direction were less pronounced, indicating the widespread acquisition of novel microbial clades in the human intestinal microbiome. A similar pattern was found for *G. b. beringei* and multiple *Pan* subspecies, however not for *G.g. gorilla*. *G.g. gorilla* and *P.t. troglodytes* are sympatric species and were sampled in the same environment in the Republic of Congo and Gabon. The absence of excess strong host-specific signals between these particular taxa might point towards an effects of a shared environment influencing microbiome structure. Human subgroups from Europe and Africa showed strong pairwise SGB-sharing between CIV and DRC, and DK and GER, respectively, however not across geographic regions, suggesting a strong connection of the microbiome with environment and lifestyle differences. All human subgroups exhibit a strong depletion of SGB-sharing with all other host genera, which resulted in

a clear separation of the human microbiota from that of other hominids.

## Fecal microbiome of European populations marked by loss of evolutionarily conserved core microbiota
Abundance difference in microbial clades between humans and NHAs and between human communities with differing environments, such as living in rural or urban regions, in regions of the world with lower or higher HDI, can give insights into microbiome-mediated adaptations to environmental changes in the distant and more recent past. We analyzed and compared the abundance profiles of gut microbes between NHAs and humans, including individuals living in rural, lower HDI areas of Africa (CIV and DRC) as well as individuals residing in urban, higher HDI regions within Europe (GER and DK). For all following taxonomic and functional comparisons, we restricted the analysis to human and *Pan* (chimpanzees and bonobos) samples to obtain focused insights into the microbiota divergence since their hosts diverged about 7-8 million years ago[25].

A total of 310 microbial genera were included in the analysis, of which 173 were found to be differentially abundant (Q$_{Bonferroni}$<0.05, Fig. 2a, Suppl. Data 5) in at least one of these comparisons between human subgroups, or between humans and NHAs, and subsequently sorted into one of four groups. We identified 57 taxa with increased abundances among humans from high HDI regions in Europe, such as *Akkermansia*, *Bacteroides*, and *Alistipes* (Fig. 2a, b). We additionally

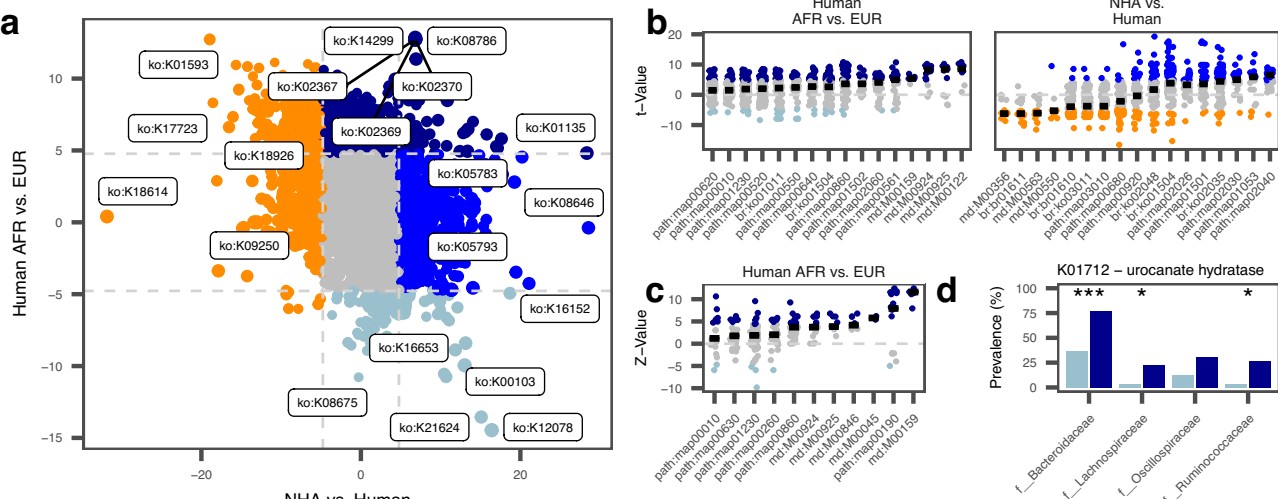

**Fig. 3 | Functional differences in the microbiota of humans and NHAs. a** Effect sizes (t-values from univariate linear regression) of the abundance differences of KEGG orthologs in fecal samples of NHAs ($n = 143$) and humans ($n = 47$, x-axis) and humans within African ($n = 23$) and European ($n = 48$) populations separated as well (y-axis). Points are colored according to the association groups with European (dark blue) and African human populations (light blue), or according to general enrichment in NHAs (orange) or humans (mid-blue). Taxonomic groups not found associated with any of the groups (all $Q > 0.05$, two-sided) are shown in grey. Horizontal and vertical lines depict the t-value threshold ($|$t-value$| > 4.77$) for statistical significance after Bonferroni-correction. **b** KEGG ortholog (KO) effect sizes from the previous analysis for differential abundance between African vs. European human population and NHA vs. Human associated taxa, respectively. Shown KOs are ordered into functional higher-level KEGG categories that were found enriched ($Q_{Fisher's} < 0.05$, two-sided) among KOs with significantly different abundances between groups. Horizontal bars indicate median t-values of all KOs in a KEGG category as an estimate for the direction of the enrichment. **c** KOs found enriched in the pangenomes of humans from Africa or Europe across four bacterial families shared across continents. Shown are the Z-values of the fixed-effects meta-analysis of KOs for enrichment across microbial families. KOs are sorted into functional higher-level KEGG categories that were found enriched ($Q_{Fisher} < 0.05$, two-sided) among KOs with significantly different prevalence ($Q_{Meta} < 0.05$, two-sided) between African and European pangenomes. Horizontal bars depict median $Z_{Meta}$-values of KOs within higher-level KEGG categories. **d** Prevalence of the urocanate hydratase gene (K01712) in clades found higher abundant in humans from Europe across four microbial families. *P*-values (Fisher's exact test, two-sided): $P_{Bacteroidaceae} = 4.55 \times 10^{-7}$, $P_{Lachnospiraceae} = 0.017$, $P_{Oscillospiraceae} = 0.052$, $P_{Ruminococcaceae} = 0.030$. Stars indicate per-clade differences in gene prevalence (Fisher's exact test, two-sided): *$P < 0.05$, **$P < 0.01$, ***$P < 0.001$.

found 39 taxa that are enriched in the two African populations such as *Cryptobacteroides, Prevotella*, and *Succinivibrio* (Fig. 2a, b). Overall, the marker taxa of European microbiomes that we detected are in agreement with previous findings[26,27]. Our approach allowed us to identify bacterial taxa that exhibit differential abundance profiles between humans and NHAs that are independent from the human populations (Fig. 2b). We found 74 taxa, such as *SIG603* that have increased abundance in NHAs and 50 taxa with increased abundance in humans, of which 15 do not show an association to either of the human subgroups, such as *Coprococcus* and *Agathobacter*. Interestingly, taxa depleted in the microbiome of European individuals compared to humans from Africa are more likely to be also abundant (>0.1%) in the microbiomes of NHAs ($P < 0.001$; Fig. 2c), suggesting a loss of evolutionary conserved clades in these populations.

### Widespread changes in fecal microbiome function between hosts and across human communities

Taxon-specific changes reflect broad-scale differences between host-groups. A focused analysis of microbial functions can give insights into the specific driving forces of such community-level changes. We performed analysis of abundance differences of 6,340 KEGG orthologs (KOs[28],) in NHA (genus *Pan*) vs. human fecal microbiota and humans in European and African societies and found significant abundance differences in 1,092 (17.2 %) and 881 (13.9 %) KOs, respectively ($Q_{Bonferroni} < 0.05$; Fig. 3a, Suppl. Data 6).

Analysis of higher-level KEGG annotations, including e.g. complete pathways overrepresented among differentially abundant KOs, revealed seven annotations with enrichment of KOs higher abundant in NHAs compared to humans and nine annotations conversely enriched in humans. Sulfur metabolism (map00920) was generally overrepresented in differentially abundant KOs with individual effect

directions associating with both groups ($Q_{Bonferroni} < 0.05$; Fig. 3b, Suppl. Data 7). NHA associated categories md:M00356, md:M00563 (methanogenesis) and path:map00680 (methane metabolism) clearly indicate a higher abundance of methanogenic archaea. Additionally, we find multiple categories involving ribosomes, including distinct ribosome annotations of archaea, which we could confirm using the pangenome distribution of these KOs (e.g. 92.9% of SGBs with K02866 [large subunit ribosomal protein L10e] belonging to the domain Archaea). In humans, we find enrichments in categories covering antimicrobial resistance (path:map01501, br:ko01504), bacterial mobility (path:map02030, path:map02040, br:ko02035), biofilm formation (path:map02026) and prokaryotic defense systems (br:ko02048), suggesting a generally higher abundance of virulence genes in human microbiomes, independent of geography. Pangenome distribution of the significantly different KOs additionally confirm, these are not driven by specific microbial clades since they are present across the entire phylogeny.

Enrichments of higher-level KEGG annotations between humans from Africa and Europe were exclusively found for the European subgroup ($n = 16$). These involved vitamin B$_{12}$ / cobalamin biosynthesis (md:M00122, M00924, M00925, and path:map00860) and antimicrobial resistance genes (br:ko01504, path:map01502). Increased antimicrobial functions may be explained by a higher use of antibiotics in human healthcare and extensive animal husbandry, as well as by environmental pollution[8]. The increased abundance of vitamin B$_{12}$-producing microorganisms in the feces of Europeans may be driven by higher dietary intake of meat and dairy products from ruminants, as these food groups contain microorganisms with this metabolic capacity[29]. Additionally, we find multiple categories suggesting a community-level shift towards oxidative carbohydrate metabolism, e.g. glycolysis/gluconeogenesis (path:map00010), pyruvate

**Table 1 | KEGG functional groups with significant enrichment ($Q < 0.05$, two-sided) in the gut microbiome of humans living in Europe**

| KEGG ID | Name | # of KOs | Mean Z-Score | P-value | Q-value |
|---|---|---|---|---|---|
| md:M00159 | V/A-type ATPase, prokaryotes | 9 | 11.19 | 3.23E-16 | 2.64E-13 |
| path:map00190 | Oxidative phosphorylation | 223 | 5.83 | 1.39E-08 | 1.13E-05 |
| md:M00924 | Cobalamin biosynthesis, anaerobic, uroporphyrinogen III => sirohydrochlorin => cobyrinate a,c-diamide | 22 | 3.59 | 2.38E-08 | 1.94E-05 |
| path:map00860 | Porphyrin metabolism | 139 | 3.51 | 5.08E-08 | 4.15E-05 |
| md:M00045 | Histidine degradation, histidine => N-formiminoglutamate => glutamate | 8 | 5.79 | 4.37E-07 | 3.57E-04 |
| path:map00010 | Glycolysis / Gluconeogenesis | 106 | 1.81 | 5.21E-06 | 4.25E-03 |
| md:M00846 | Siroheme biosynthesis, glutamyl-tRNA => siroheme | 16 | 4.64 | 5.99E-06 | 4.90E-03 |
| md:M00925 | Cobalamin biosynthesis, aerobic, uroporphyrinogen III => precorrin 2 => cobyrinate a,c-diamide | 17 | 4.36 | 1.07E-05 | 8.78E-03 |
| path:map00630 | Glyoxylate and dicarboxylate metabolism | 101 | 1.94 | 3.60E-05 | 2.94E-02 |
| path:map00260 | Glycine, serine and threonine metabolism | 109 | 1.74 | 5.87E-05 | 4.80E-02 |
| path:map01230 | Biosynthesis of amino acids | 238 | 1.41 | 6.05E-05 | 4.94E-02 |

metabolism (path:map00620), phosphotransferase system (path:map02060), and V/A-type ATPase (md:M00159).

## Success of taxa associated to European populations relates to oxidative carbohydrate metabolism

Community-level changes in abundances of functional groups give insights into some aspects of adaptations, however, they are over-proportionally driven by highly abundant taxonomic groups. We applied pangenome analysis to specifically identify individual genes and functions enriched or depleted within specific microbial clades overrepresented the gut of human individuals residing within Europe in comparison to other human associated taxa. Analyzes were restricted to four bacterial families, *Bacteroidaceae, Lachnospiraceae, Oscillospiraceae*, and *Ruminococcaceae*, for which sufficient numbers of SGBs ($n >$ =10 in each of both groups) for pangenome analysis were recovered. SGBs in these family represent large proportions of the overall human microbiome (mean = 53.2%), with no significant differences between the subgroups ($P_{Kruskal-Wallis} = 0.2$). The analysis was conducted at the family-level, as higher taxonomic ranks would increase clade-specific functional biases. To account for between-family functional differences, functional differences of SGBs associated with Europeans were first analyzed within microbial families using Fisher's exact test and subsequently subjected to unweighted meta-analysis using Z-scores to leverage shared signals.

We identified 167 enriched and 30 depleted KOs in pangenomes associated with the European populations ($Q_{Meta,Bonferroni} < 0.05$; Suppl. Data 8 and 9). Using higher-level KEGG annotations (modules, pathways, BRITE hierarchies), we found that 11 of these annotations were overrepresented in the dataset ($Q_{Bonferroni} < 0.05$, Fig. 3c, Table 1, Suppl. Data 10). Among these were multiple groups involved in carbohydrate metabolism enriched in taxa associated with Europeans, specifically pointing at aerobic breakdown of sugar molecules for ATP generation (citrate cycle: md:M00009, path:map00020; pentose phosphate pathway: md:M00004; V/A-type ATPase, prokaryote: md:M00159), confirming patterns seen also in the community-level analysis of functional abundances. These signals strongly suggest that the selection for taxa in the gut of humans living within Europe is connected to a diet rich in carbohydrates and potentially the adaptation to transient microaerobic conditions in the gut environment, using oxidative phosphorylation as a mean to release energy from nutrients, which is more efficient than strictly anaerobic fermentation[30]. However, reduced fermentation can impact short-chain fatty acid production, which can, in turn, potentially negatively affect the host's intestinal epithelial cells and metabolism[31]. Additionally, we found a pathway connected to the histidine degradation (md:M00045). Especially gut microbial histidine metabolism has been

discussed with relevance to human health, as it was shown that an intermediary product of histidine degradation, imidazole propionate, was increased in type 2 diabetic individuals in a large study of almost 2000 individuals, and that this increase was directly connected to the microbiota and overall unhealthy dietary habits, however independent of dietary histidine intake[32]. The enzyme urocanate hydratase (EC:4.2.1.49; K01712) is responsible for the interconversion between urocanate and imidazole propionate in the histidine degradation pathway. We find this gene significantly enriched in three of the four bacterial families analyzed ($P < 0.05$, Fig. 3d), with a clear trend visible also for *Oscillospiraceae* ($P = 0.052$). This suggests a microbiome-encoded pre-disposition to metabolic disorders in European human communities.

We did not find any higher-level KEGG annotations significantly enriched among the taxa not enriched in Europeans. The relative lower fitness of these taxa may not result from a single mechanism associated to European lifestyles, but rather from multiple selective forces specific to the individual clades.

## Convergent host-specific adaptations are found across microbial families

Shared gene gains or losses across multiple microbial clades can indicate a response to specific host intestinal environments, leading to functions being acquired (and selected for) multiple times independently. We performed a pangenome analysis of genera shared between humans and NHAs ($n = 36$) to identify such patterns of convergent adaptation. To control for higher-level clade effects, functional repertoires (KEGG Ontology terms) were compared between SGB pangenomes at the genus-level from NHAs (Suppl. Data 11) and humans; results were then combined in a meta-analysis across genera. In total, 78 KO terms were identified as carrying signatures of convergent adaptation to the respective host group, with 57 of these signatures associating with humans and 21 associating with NHPs ($Q_{Bonferroni} < 0.05$; Fig. 4a, Suppl. Data 12). Among the human-associated KOs, we found multiple functional groups hinting again at an adaptation to increased oxygen by utilization of oxygen as an electron acceptor within the respiratory chain, such as cytochrome *bd* ubiquinol oxidase subunits (cydA, cydB), as well as adaptation to increased oxidative stress through ferritin (ftnA) and thioredoxin-dependent peroxiredoxin (BCP). Among the KOs enriched in NHP pangenomes we found an outer membrane factor (TC.OMF), a major facilitator super-family (MFS) transporter (lrmB), 1-epi-valienol-7-phosphate kinase (acbU), and two KOs annotated as polyketide synthases (rhiA, pksN). Products produced by polyketide synthases have diverse functions, including antibiotic activity, virulence, and support of symbiotic relationships[33]. OMFs and MFS form transmembrane complexes for

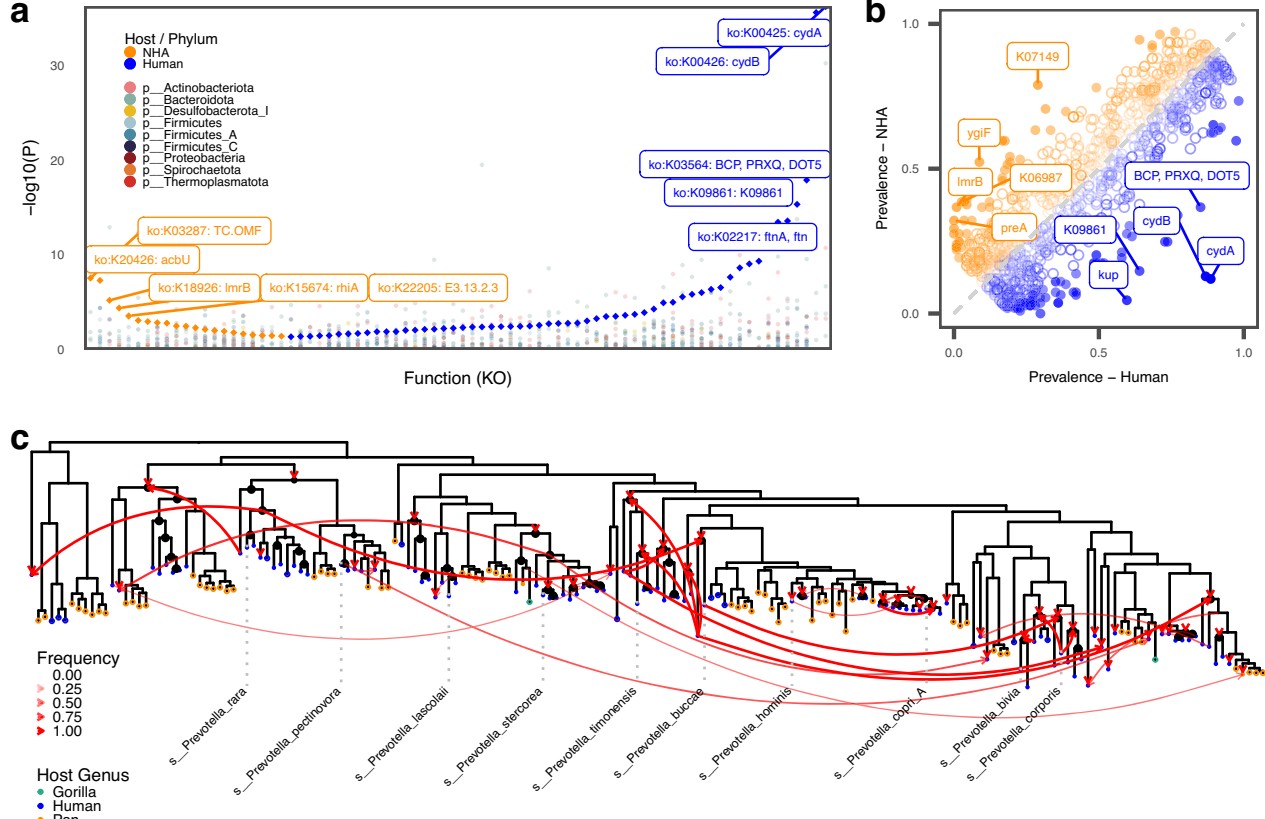

**Fig. 4 | Cross-microbial clade functional associations with NHA and human hosts. a** KO terms consistently enriched in human- (blue diamonds) and NHA-associated (orange diamonds) taxonomic clades. Individual genus-level $P_{Fisher}$-values are shown in points colored by phylum. Shown *P*-values are unadjusted for taxon-level tests and Bonferroni-corrected for multiple testing for the meta-analysis. All tests were two-sided. **b** Prevalence patterns of KO terms in human- (blue) and NHA-associated (orange) *Prevotella* SGBs. Filled shapes represent KOs with significant (Q < 0.05, Fisher's exact test, two-sided) differences in the statistical test. **c** Results of the tree reconciliation analysis for the cydA gene in *Prevotella* SGBs found in humans (blue) and NHAs (*Pan*: orange; *Gorilla*: green) demonstrate a history of frequent transfer events across 1000 reconciliations with random seeds. Filled and empty shapes represent cydA-positive and -negative SGBs, respectively. Red arrows depict gene transfer events from a donor to a recipient node found in at least 50% of reconciliations and are weighted by frequency. Red triangles mark nodes that were identified as gene transfer recipients with >50% frequency independent of the donor node. Black circles mark speciation events with >50% frequency. The ten highest abundant *Prevotella* species with established names are shown for orientation. The Prevotella tree was rooted using *Paraprevotella clara* as the outgroup (not shown).

the transport of a large variety of solutes[34], including e.g. carbohydrates, metal ions, amino acids, and export of toxic compounds[35,36]. How these potential adaptations relate to NHA hosts is unclear, however, they might indicate adaptation to a diverse diet[37,38] or metabolism of plant-derived xenobiotics[39].

*Prevotella* represented the largest genus-level clade in the dataset (*n* = 212, 3.7% of all SGBs). While we found this genus across all host species, it is largely decreased in abundance and prevalence in Europeans. We selected this genus for further analysis to elucidate potential functional mechanisms driving the observed patterns. Enrichment analysis revealed 126 KOs with distinct prevalence patterns (Q < 0.05; Fig. 4b). The most striking difference was found for the cytochrome *bd* ubiquinol oxidases subunits 1 and 2 (cydA and cydB), which were found in 101 of 114 human-associated *Prevotella* SGBs (incl. UHGGv2 genomes), but present in only two out of 72 SGBs from NHAs. It is important to note that these prevalence differences are not driven by single lineages within the *Prevotella* genus that would have distribution ranges restricted to single hosts. Instead, they are observed across multiple sibling clades spanning the entire phylogenetic tree of the taxon (Fig. 4c). Cytochrome *bd* oxidases are involved in stress responses, most prominently in transiently microaerobic environments[40]. Comparison of the *Prevotella* species tree (reconstructed using all SGB representatives recovered from the dataset (*n* = 184)) and the cydA gene

phylogeny exhibit widespread incongruencies between their tree topologies (Fig. 4c). We performed a tree reconciliation using a duplication-transfer-loss (DTL[41];) model between the *Prevotella* and cydA phylogenies, which revealed frequent events of gene transfer ($\bar{T} = 44.6$) between branches (including distant ones) of the *Prevotella* phylogeny and subsequent losses ($\bar{L} = 23.2$) in the NHA-associated clades. Standardized by gene prevalence, these values are in the 71st and 61st percentile for transfers and losses of single-copy genes found in the *Prevotella* genus, respectively (Suppl. Data 13). Interestingly, the two cydA-carrying SGBs found in NHAs are phylogenetically distant, however, their cydA genes are highly similar and most likely the result of a transfer from one to the other. The gene transfer events and mappings were robust across 1000 reconciliations with different starting seeds, with 89.2% of all events and 67.6% of all mappings found with 100% consistency. These results suggest that the enrichment of cytochrome *bd* ubiquinol oxidases observed in humans compared to NHAs are the result of multiple reoccurring events of gene loss and horizontal gene transfer between *Prevotella* clades within the hominid gut.

## Co-phylogeny is associated with enrichment of microbial traits and disrupted in humans

Patterns of co-phylogeny between host and microbes can result from close interaction, or even interdependence in extreme cases, and

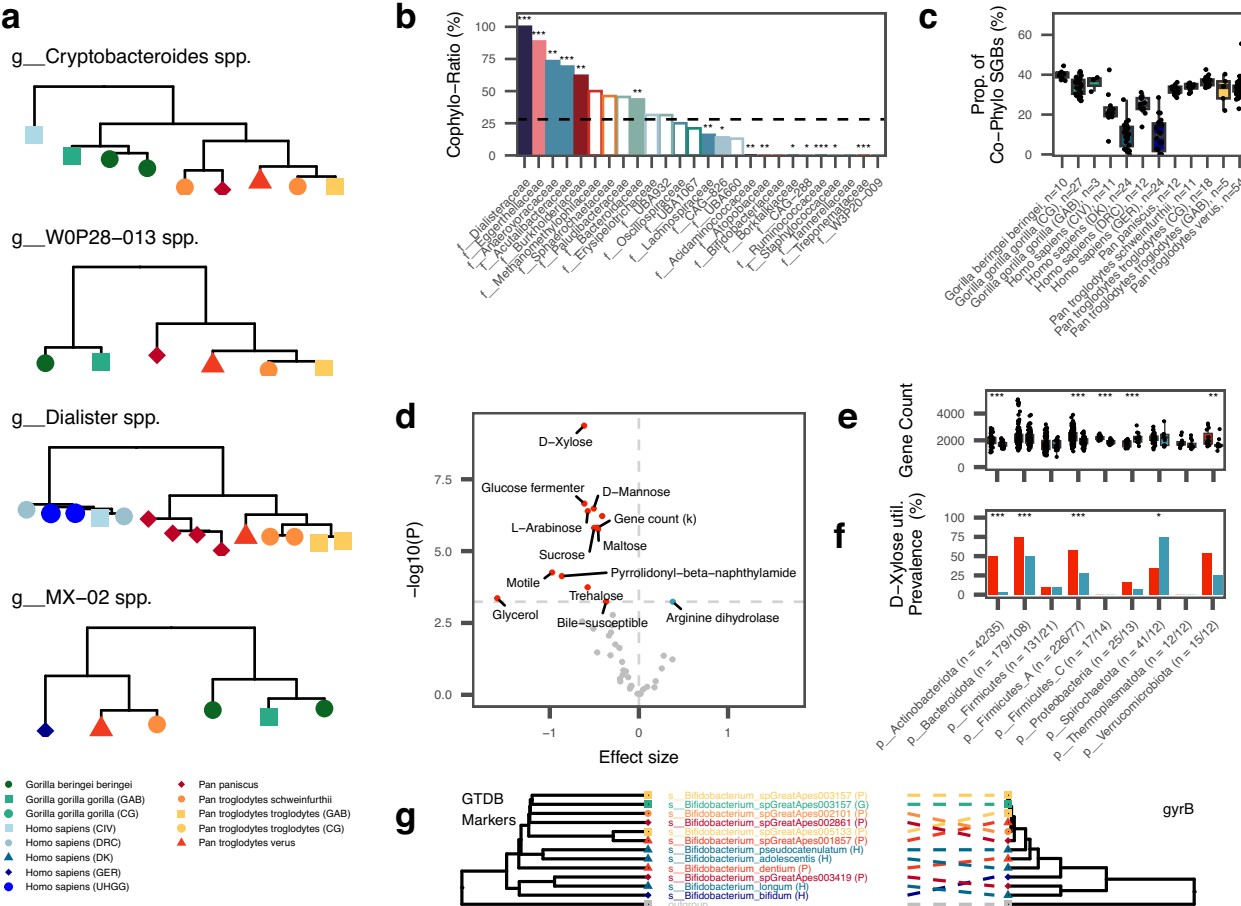

**Fig. 5 | Cophylogeny across humans and non-human African great apes.**
**a** Subtree phylogenies of groups with significant results in the Mantel-test based analysis for co-phylogeny. Tip colors and shapes correspond to the host subgroups. Trees were rooted on a randomly selected outgroup from a related family (not shown). **b** Enrichment and depletion of co-phylogeny patterns across microbial families with at least 10 SGBs in the analysis. Bars are colored by phylum, corresponding to the colors in Fig. 1. Filled bars denote significant ($Q_{Fisher}<0.05$, two-sided) enrichment and depletion. The dashed line represents the average co-phylogeny ratio across all SGBs. **c** Per-sample proportion of SGBs with co-phylogeny patterns across and colored by host subgroups. Group sizes are given in the x-axis labels. **d** Enrichment (blue) and depletion (red) of 43 in-silico inferred microbial traits, genome size and gene count in association with cophylogeny signals. Effect sizes and $P$-values from mixed effects logistic regression accounting for phylogenetic relatedness of SGBs. The horizontal line marks the threshold of

significant Bonferroni-adjusted $P$-values (two-sided). **e** SGB-level gene counts across nine phyla, grouped by the presence (blue) and absence (red) of a cophylogeny signal. Within-phylum differences were assessed by two-sided Wilcoxon rank-sum test. **f** Prevalence of inferred D-Xylose utilization by SGBs across phyla, grouped by the presence (blue) and absence (red) of a cophylogeny signal. Within-phylum differences in prevalence were assessed using a two-sided Fisher-test. Group sizes of SGBs within phyla negative and positive for cophylogeny signal are given in the x-axis labels ($n$ = neg/pos). **g** Tanglegram of Bifidobacterium maximum-likelihood phylogenies based on 120 GTDB marker genes (left) and gyrB sequence (right). Tip colors and shapes correspond to the host subgroups. Across all panels, stars indicate level of significance: * $P<0.05$, ** $P<0.01$, *** $P<0.001$. Exact $P$-values can be obtained from Supplementary Data 14–16. All boxplots show the following elements: center line: median, box limits: upper and lower quartile; whiskers: 1.5 × interquartile ranges.

congruent metabolic pathways from co-evolutionary trajectories. Using stringent selection criteria, we subjected 209 subtrees of the SGB phylogeny for co-phylogeny analysis (see Methods). The subtrees spanned 46 families and 945 (28.8%) SGBs present in the dataset in addition to 77 SGBs from the UHGGv2 catalog. We used a Mantel-test based framework and permutation to detect co-phylogeny signals[42], (see Methods for details). When defining co-phylogeny candidates based on a mean $P$-value < 0.05 across all permutations, 56 of 209 subtrees (26.8%) qualified as exhibiting co-phylogenetic patterns (Suppl. Data 14). These subtrees cover 312 of the 1051 SGBs (30.7%) included in the analysis and 5.84% of the total 5345 hominid SGBs (excluding the SGBs from Manara et al.). All results and subtrees can be inspected online (https://mruehlemann.shinyapps.io/great_apes_shiny_app). By visually inspecting subtrees with co-phylogeny signals, we find many candidates microbial phylogenies that do not follow host phylogeny, e.g., in a subgroup of the genus *Cryptobacteroides* (Fig. 5a). Such signals suggest that co-phylogeny within the *Gorilla* and

*Pan* clades can result in statistically significant Mantel tests, despite topological incongruences of human-derived sequences, for which no host sister (sub-)species from the same genus is available.

Human-derived genomes were found in 149 (71.3%) of the tested subtrees, of which 38 (25.5%) were co-phylogeny candidates. Similarly, 18 of 60 (30%) tested subtrees without human-derived representatives exhibited co-phylogeny signals, *e.g.*, *WOP29-013* spp. (Fig. 5a). Overall, 21.8% ($n$ = 82 out of 377) of human-derived SGBs in the analysis were found in co-phylogenetic subtrees, which is significantly less compared to 35.7% ($n$ = 231 out of 647) of NHA-derived SGBs ($P_{Fisher}$ = 2 × $10^{-6}$). Co-phylogeny signals, defined based on the ratio of SGBs in trees exhibiting co-phylogeny patterns and the total SGBs in the family included in the analysis, differed strongly. Six families showed excessive signals of co-phylogeny and nine families a significant depletion ($Q_{Fisher,\ FDR}$ < 0.05; Fig. 5b, Suppl. Data 15). The highest co-phylogeny ratio was found for the family *Dialisteraceae* (Cophylo-Ratio = 100%, Q = 4.61 × $10^{-7}$), which in the analysis were entirely represented by 14

SGBs in the genus *Dialister*. *Dialister* are a common, but rather neglected members of the gut microbiota which have been found increased[43,44] and decreased[45] with various human diseases, hence their relation to human health remains unclear. However, on species, *Dialister invisus*, was found to be moderately transmissible between human mother-infant pairs and within households in a large meta-analysis[46]. The strongest depletions were found for e.g. the families *Lachnospiraceae* and *Treponemataceae*, the former confirming previous results for this clade[13]. The latter, *Treponemataceae*, especially the genus *Treponema D*, were found depleted in humans living in Europe and occur in anaerobic sediments[47], serving as (intermediary) reservoirs for transmission to humans and NHPs, which can disrupt co-phylogenetic signals by constant re-introduction to the community.

When comparing host groups, the proportion of SGBs with co-phylogeny signal is significantly reduced in all human subgroups compared to the NHA hosts (all $Q_{Wilcoxon} < 0.05$; Fig. 5c). Further, humans from Germany and Denmark exhibited even lower proportions of co-phylogeny SGBs compared to the two African human populations ($Q_{Wilcoxon} < 0.05$), but not to each other ($P_{Wilcoxon} = 0.68$). The human subgroups from Africa did not differ in their co-phylogeny proportions ($P_{Wilcoxon} = 0.096$). These results suggest a loss of wild great ape-associated clades in the intestinal microbiota of humans independent of their geographic origin and the introduction of novel microbial partners with changing environments and lifestyle, confirming again findings from previous analyzes.

Signals of cophylogeny suggest a strong association and possible adaptation with the host and the reduction of genome size and gene content are expected patterns connected to this process[48] which has previously been shown also for microbes with codivergence patterns in human population[11]. To explore whether these processes could be detected in our dataset, we analyzed genome size and gene count, as well as 43 microbial traits inferred from genome-level annotations for signals in association with co-phylogeny using logistic regression, while controlling for phylogenetic relatedness (see Methods; Suppl. Data 16). Out of 45 analyzed traits, 13 were found significantly depleted in clades exhibiting co-phylogeny signals ($Q_{Bonferroni} < 0.05$; Fig. 5d; Suppl. Data 17), including genome size, gene count, capabilities to use multiple simple sugars, and bile susceptibility. Phylum-level analyzes confirm this overall trend especially for Actinobacteriota, Firmicutes A, Firmicutes C and Verrucomicrobiota ($P_{Wilcoxon} < 0.05$), while Proteobacteria exhibited an inverse signal (Fig. 5d). This inverse signature might be explained by host-specific acquisition of genes found e.g. for *E. coli* across different and diverse hosts[49], however, these findings need to be further investigated. A higher susceptibility to bile in clades without co-phylogeny signal clearly suggests that bile tolerance is an adaptation to the Intestinal environment of hominid hosts. The depletion of the utilization of simple dietary sugars (traits: D-Xylose (Fig. 5f), D-Mannose, Maltose, Glucose fermenter, Sucrose, Trehalose, L-Arabinose; all $Q < 0.05$) in co-phylogeny clades might suggest an adaptation to host-derived complex carbohydrates or other energy sources in the host-context.

Only one trait – the Arginie dihydolase pathway, also called Arginine deiminase pathway (ADI) – was found enriched with co-phylogeny signal. Argrinie metabolism has been widely discussed in the context of host-microbe interaction[50], and ADI specifically was shown to protect bacteria from acid stress in host-association[51], but was also shown to modulate host immunity[52] and as specifically acquired by Saccharibacteria in the process of colonization of mammals[53], confirming its potential association with co-phylogeny in hominids. Previous analyzes suggested that spore-forming clades are less likely to be exhibiting co-phylogenic patterns, due to their ability to survive outside of the gut, facilitating dispersal between hosts[10,13,54,55]. We did not find any negative correlation between spore-formation ability and co-phylogeny patterns ($Q = 1$, Suppl. Data 16), however the annotation of this trait was restricted to only two phyla

(Firmicutes and Firmicutes A) and was also rare within these clades, being found in only 54 SGBs across the dataset. As such, whether these results contradict previous findings cannot be concluded in this analysis and warrants future focused analyzes.

We found 67 out of 157 SGBs in *Bacteroidaceae* within subtrees with cophylogeny signals (Co-pylogeny-Ratio = 43.5%), consistent with previous findings based on *gyrB* amplicon data[13]. However, no evidence for strict co-phylogeny was found in *Bifidobacteriaceae* ($n_{SGB}$=10 in the analysis, none with co-pyhlogeny signals), which is inconsistent with findings from the same report[13]. Comparing the phylogeny of metagenome-derived *gyrB* sequences and the GTDB marker-gene phylogeny for *Bifidobacterium spp.* revealed clear incongruences between both approaches (Fig. 5g), which may explain the differences in the presented analysis and previous findings.

## Discussion

Here, we present the largest curated dataset of fecal metagenomes derived from wild African great apes and human populations. For this, we surveyed and reconstructed high-quality microbial genomes from the feces wild non-human apes, including gorillas (*Gorilla gorilla gorilla*; *Gorilla beringei beringei*), chimpanzees (*Pan troglodytes verus*; *P.t. troglodytes*; *P.t. schweinfurthii*), and bonobos (*Pan paniscus*) as well as human populations from Africa and Europe. We identified signals of phylosymbiosis across the included hominids, indicating a conserved evolutionary relationship of microbial communities with their host species. Moreover, by employing a comparative approach, we found extensive changes of microbial taxonomic and functional abundances across the intestinal microbiota of NHAs and humans. Previous studies have pointed to "Western" lifestyles as an important factor influencing the intestinal microbiota in humans. Within our human sample population, we were able to confirm differential signals of prokaryotes associated with the European human populations. Importantly, using a comparative dataset of great ape taxa showed that microbial clades lost in Europeans in comparison to African human populations are also found in wild great ape populations. Thus, we suggest that the loss of these taxa might be regarded as the departure from a natural divergence trajectory since their last shared ancestor, cumulating in a mass extinction event of evolutionary conserved members of hominid-associated gut microbiota. While it is tempting to link these changes to industrialization (as previous studies have done), there are many differences between these human populations (e.g., genomic diversity, diet, exercise, sunlight exposure, exposure to antibiotics, population bottlenecks) and it was certainly not possible with the sampling regime here, to determine the particular factors responsible for the variation observed between the human populations sampled here. Due to logistic constraints, preservation methods for fecal samples from the included hosts and host subgroups differ. While these are expected to influence microbiome composition, previous studies show that individual signatures are retained independent of storage methods[56]. Despite this caveat, the fact that considerable variation exists between human populations is notable and highlights the need for much higher-resolution sampling of human-associated microbial diversity. Similarly, our analysis suggests that there is even more undescribed microbial diversity to be discovered across populations of wild non-human apes.

In a pangenome analysis to identify individual genes or functions enriched or depleted in genomes of taxa associated with different human populations, we identified numerous functional traits involved in aerobic respiration associated with the European populations in the analysis. We hypothesize that taxa found to be enriched in the fecal samples of humans from Germany and Denmark might have a selective advantage via their clade-independent ability to survive or even utilize aerobic conditions in the intestinal tract. More specifically, we propose that these taxa have undergone convergent adaptation to tolerate high oxygen concentrations. Such aerotolerance could increase microbial

fitness, whereby bacteria can withstand high oxygen concentrations to metabolize mucus layers for energy[57]. However, the depletion of this mucus by bacteria diminishes an important physical and immunological barrier that protects the human host against microbial assaults and allows for direct interaction between host epithelial cells and microbiota, potentially triggering (auto-)immune processes[58,59]. Notably, we showed that the introduction of novel microbes associated with industrialization related to vast differences in the community composition of fecal microbiota in the European populations. Thus, it is possible that susceptibility to intestinal inflammation might be potentiated by specific taxa found in this population. Accordingly, we found increased abundances of well-characterized mucin-degrading taxa, including *Akkermansia* and *Bacteroides*, in the European cohorts. These findings are congruent with previous reports suggesting that there is increased mucus degradation by intestinal microbiota in human populations with direct access to industrial food systems, which may relate to higher incidences of inflammatory bowels diseases observed in developed economies[6].

Comparatively, only a few pathways showed conserved enrichment in the opposite direction, suggesting clade-specific mechanisms for their loss in some human societies. In particular, we found the taxon *Prevotella* is depleted in German and Danish samples but conserved across hominids, despite representing a diversity of host clades and diets. *Prevotella* is a major determining taxon of one of the human enterotypes, a concept used to define fecal microbial communities[60]. It remains controversial as to whether the human gut microbiome is best classified using such discrete categories, or rather along a dynamic, continuous gradient (Cheng 2019, Knight 2014). Nevertheless, previous reports have shown that individuals who access industrialized food systems (i.e., consume so-called "Westernized" diets) generally display a *Bacteroides*-dominant enterotype. *Bacteroides*-enterotypes have been previously associated with a multitude of intestinal[61] and extra-intestinal inflammatory diseases[62]. Conversely, individuals who rely on rural and traditional subsistence strategies (i.e., consume plant-rich diets) tend to exhibit a *Prevotella*-dominant enterotype[3]. This enterotype is also displayed in about 20% of individuals living within Western societies[63]. Interestingly, there are conflicting reports concerning *Prevotella* and host health. While it has been shown that *Prevotella* may improve glucose metabolism[64], other reports have linked high abundances of *Prevotella* spp. with autoimmune diseases and intestinal inflammation[65]. While results from model systems have suggested *Prevotella* likely plays a role in autoimmunity, these studies largely relied on mono-colonization of germ-free animals and thus may be biased due to a lack of microbial interaction partners and an aberrant host physiology[65]. Within human studies, no convincing link between increased *Prevotella* spp. and inflammatory bowel disease has yet been shown[65].

Here, we used an evolutionarily informed framework to extend the enterotype concept to elucidate the functional dynamics involved in the assembly of the human gut microbiome over evolutionary timescales. Such insights may better inform how changes in the gut microbiome might affect human health. We find the taxon *Prevotella* to be conserved across the sampled hominids. Moreover, the sheer diversity of *Prevotella* displayed across all hominid clades clearly suggests evolutionary conservation and long-standing interaction of this microbial clade with the host, as further revealed by host-specific microbial functions identified in the metagenomic pangenome analysis. In other words, we find the *Prevotella* clade to be an integral member of the intestinal microbial community of all hominids. Therefore, we propose that the *Prevotella*-enterotype represents an evolutionary ancestral community state for the human gut microbiome. Rather than a discrete enterotype, a reduced abundance and diversity of *Prevotella* may better regarded a key biomarker for disease risk[66] or for microbiota insufficiency syndrome[67], which seems to be partly driven by changes associated with Western lifestyles. Additional

research and large-scale strain collections for *Prevotella* are needed for in-depth analysis and evaluation of this diverse taxonomic group with regard to host health and its role in inflammation. Such research must consider *Prevotella spp.* as members of a complex consortium of interacting microorganisms and as, we argue, a potential target for pre- and probiotic intervention in chronic inflammatory disorders.

Lastly, we leveraged our catalog of high-quality metagenome-assembled genomes from hominid fecal samples together with existing data to investigate co-phylogenetic patterns across the sampled hosts. Overall, co-phylogeny showed highly clade-specific enrichments and depletions. In addition, human-derived MAGs were found significantly less often among co-phylogenetic groups than MAGs from NHAs. Since we included human-derived data from global reference datasets[24], this effect is unlikely to be an artifact of non-exhaustive coverage of human microbiome members. We found several microbial traits depleted among bacteria with cophylogenetic patterns. Among these were, as expected, reduced genome sizes and gene counts, as well as susceptibility to bile and utilization of multiple simple carbohydrates. We hypothesize that these depletions mirror the specialization of microbes to colonize the hominid gut and utilize host-derived complex carbohydrates.

Our study has limitations. The co-phylogeny analysis relies heavily on genome-sequences recovered from shotgun metagenomic sequencing (MAGs), which are potentially contaminated and incomplete, which could bias tree structure and thus, co-phylogeny estimates, and also can potentially under- (or over-)estimate the functional capacities of recovered microbial genomes. To address the potential shortcomings of MAGs, we implemented stringent data processing pipelines and quality control to achieve high-quality MAGs and a host-specific pan-genome-based functional annotation framework incorporating information from multiple MAGs per species representative to reduce potential genome gaps (see Methods). Additionally, the commonly used estimates of divergence times of the hominid hosts included in the analysis set the timeframe of the split from a shared ancestor to 8–19 million years ago[25,68]. Bacterial speciation events happen in the timeframe of 10–100 million years, or longer[69–71], and thus, co-phylogeny in hominids is expected to be observed within microbial species or possibly genera. In the presented dataset, species-level sharing of MAGs between host genera was low (1.68%; $n = 30$ out of 1,787 reconstructed SGBs, not including Manara et al. and UHGGv2). SGBs were defined on 95% average nucleotide identity, a measure generally regarded as appropriate[72], but it is nevertheless prone to clade specific biases, potentially further influenced by altered speciation dynamics in association with (evolutionary) changes in host lifestyle[73], *i.e.* previously demonstrated increases in horizontal gene transfer (HGT) within individual microbiomes[8]. Accounting for these potential biases, we relaxed the threshold to shared genus-level annotations for subgroups to be included in the co-phylogeny testing, while keeping the number of tested sub-phylogenies to a minimum through the definition of stringent inclusion criteria (see Methods). Despite these considerations, the observation of signals of co-phylogeny across hominids is supported by a robust statistical framework.

Additional limitations stem from the focus on humans and African great apes. While the comparisons between these host-clades provide a framework for the in-depth investigation of (evolutionary) rather recent adaptations and between-host divergences, they potentially neglect that could be revealed by broader-scale investigations, such as the previously described convergence of the human gut microbiota towards that of cercopithecines[17]. However, our analyzes show the impact of the unique trajectories taken by the intestinal microbiomes of *Pan* and *Homo* since their last common ancestor.

Our work here lays the foundation for the analysis of disease-associated changes in the human intestinal microbiome in an evolutionarily informed framework, thereby allowing researchers to

evaluate microbiome-associated inflammatory disorders from a point of view that considers both proximal and evolutionary influences. Future investigations should consider in-depth analysis of horizontal gene transfer events within or even between primate hosts to shed further light on also cross-species dynamics and transition of microbes. Such analyzes however require either microbial isolate genomes or at least long-read sequencing data to increase confidence in detection events. Additionally, time series data for host groups sharing the same habitat, *e.g. G. g. gorilla* and *P. t. troglodytes*, could give additional insights into cross-species sharing dynamics which cannot be appropriately elucidated based on single-timepoint data.

In summary, we present an in-depth taxonomic and functional description and analysis of hominid-associated fecal communities spanning about ten million years of evolution and host-microbiome interactions in the gut of humans and African great apes. Western lifestyle and maybe more precisely industrialization-associated changes in human gut microbiota have been previously suggested as a driver of microbiome insufficiency syndrome, whereby an incompatibility between quickly adapting microbiota and slowly evolving host genes leads to chronic inflammatory diseases such as metabolic syndrome, type 2 diabetes, and inflammatory bowel disease[6,74]. Thus, a comparative analysis of human and NHA intestinal microbiota that considers evolutionary forces as presented herein provides a powerful platform to advance our understanding of human-associated microbiota and guide the development of personalized, targeted interventions to prevent and treat chronic inflammatory disorders.

## Methods

### Ethics & inclusion statement
Ethical approval for work on human samples was obtained from the Local Ethics Committee Germany, Kiel (reference number A156/03), the Ivorian ethics commission (Comité national d'éthique et de la recherche [CNER], permit number 101 10/MSHP/CNER/P) and the Congolese ethics commission (Comité d'Éthique, Ministère de l'Enseignement Supérieur et Universiaire, permit number ESO/CE/018/11). All procedures performed in studies involving human participants were in accordance with the ethical standards of the institutional and/ or national research committee and with the 1975 Helsinki Declaration and its later amendments or comparable ethical standards.

Sampling of wild-living great apes in Africa were granted and facilitated by the following organizations. Bwindi Impenetable Forest National Park, Uganda (for sampling of *Gorilla beringei beringei*): The mountain gorilla survey was conducted by the Uganda Wildlife Authority, l'Institut Congolais pour la Conservation de la Nature, the Rwanda Development Board, the International Gorilla Conservation Programme, the Max Planck Institute for Evolutionary Anthropology, Conservation Through Public Health, the Mountain Gorilla Veterinary Project, the Institute for Tropical Forest Conservation, and The Dian Fossey Gorilla Fund and was conducted in compliance with the regulations of and permission of the Uganda National Council for Science and Technology and the Uganda Wildlife Authority.

Budongo Forest Reserve, Uganda (for the sampling of *Pan troglodytes schweinfurthii*): the Uganda National Council for Science and Technology and the Uganda Wildlife Authority, with additional ethical approval given by the School of Psychology, University of St Andrews.

Kokolopori Bonobo Reserve (for the sampling of *Pan paniscus*), Democratic Republic of the Congo: the Ministere de Recherche Scientifique et Technologie, Democratic Republic of the Congo and was supported by the Vie Sauvage, the Bonobo Conservation Initiative;

Loango National Park, Gabon (for the sampling of *Gorilla gorilla gorilla* and *Pan troglodytes troglodytes*): the Agence Nationale des Parcs Nationaux, the Centre National de la Recherche Scientifique et Technique of Gabon.

Taï National Park, Côte d'Ivoire (for sampling of *Pan troglodytes verus*): the Ministère de l'Enseignement Supérieur et de la Recherche

Scientifique, the Ministère des Eaux et Fôrets in Côte d'Ivoire, and the Office Ivoirien des Parcs et Réserves. Work was supported by the Centre Suisse de Recherches Scientifiques en Côte d'Ivoire and the staff members of the Taï Chimpanzee Project and approved by the ethics council of the Max Planck Society (4.08.2014).

Researchers from CIV and DRC contributing to conducting research and fulfilling the authorship criteria were included as co-authors. Research at sites in Africa was conducted in collaboration with local partners as stated in the acknowledgements section, granted by local authorities and in agreement with local policies. Feces from wild-living, habituated animals were collected after defecation without interfering with the animals. Research at great ape sites was increasingly performed following the IUCN guidelines to minimize disease risk for great apes. We did not stratify or correct for sex or gender effects in the analysis. Our analyzes focus on the comparison of gut metagenomes from either distinct hominid species or between human subgroups from populations with differences in the human development index. We expect that the effects of sex and/or gender are negligible in this context and these factors have not been explored in the current analysis.

### Fecal Sampling, DNA extraction and data generation
Sampling of wild-living great apes and human populations in Africa were conducted at: Bwindi Impenetrable Forest National Park, Uganda (*Gorilla beringei beringi, Pan troglodytes schweinfurthii*); Kokolopori Bonobo Reserve and villages adjacent to Salonga-Sud National Park, Democratic Republic of the Congo (*Pan paniscus*, Human); Loango National Park, Gabon (*Gorilla gorilla gorilla, Pan troglodytes troglodytes*); Taï National Park and adjacent villages, Côte d'Ivoire (*Pan troglodytes verus*, Human). Sampling procedures for collecting feces from humans (*n* = 48) and wild non-human primates (*n* = 109) have been previously described[15]. Briefly, fecal samples were collected immediately after defecation, and, depending on the local infrastructure, either stored in RNAlater and frozen at −20 °C or stored in a cryotube, cooled in a thermos until return to the field laboratory, and subsequently snap frozen in liquid nitrogen. Appropriate government permits and permission to conduct research on wild primates were granted by the relevant authorities (see Acknowledgments for site-specific details). Human fecal samples from the Democratic Republic of Congo (*n* = 12) were stored in RNAlater and frozen at −20 °C. Human fecal samples from Côte d'Ivoire (*n* = 12) were stored in a cryotube, cooled in a thermos until return to the field laboratory and subsequently snap frozen in liquid nitrogen. Human fecal samples from Germany were collected at home by the participant in standard fecal collection tubes, mailed to the study center, and stored at −80 °C. DNA extraction from fecal samples was performed from 200 mg of stool transferred to 0.70 mm Garnet Bead tubes (Qiagen) with 1.1 mL ASL buffer, followed by bead beating in a SpeedMill PLUS (Analytik Jena AG) for 45 s at 50 Hz. Samples were heated to 95 °C for 5 min and centrifuged, retaining 200 µl of the supernatant for DNA extraction with the QIAamp DNA Stool Mini Kit (Qiagen) automated on a QIAcube system (Qiagen) according to the manufacturer's protocol. DNA quality was assessed by Qubit and Genomic DNA ScreenTape (Agilent). Illumina Nextera DNA Library Preparation Kit was used to construct shotgun metagenomic libraries, and subsequently sequenced with either 2 × 125 bp reads on a HiSeq 2500 platform or with 2 × 150 bp reads on a NovaSeq 4000 machine (Illumina).

### Data processing, assembly and metagenomic binning
Raw sequencing FastQ files were quality controlled and preprocessed using the BBMap software suite[75]. Host reads were removed using bbmap.sh. A masked human reference database[76] and a lenient mapping threshold of 95% identity was used to account for a broader host range to also capture host contamination from the *Pan* and *Gorilla* host. Metagenomic contigs were assembled with metaSPAdes and

contigs >= 2000 bp were retained[77]. Reads were mapped to the contigs of the respective samples using Minimap2[78], converted to BAM files with Samtools[79] and used to estimate per-contig mapping depth with the jgi_summarize_bam_contig_depths binary from the MetaBAT2 binning tool[80]. Contig binning for individual samples was performed with MetaBAT2[80], MaxBin2[81], and CONCOCT[82]. In addition, the VAMB binning tool[83] was used on a cross-mapping catalog of the merged contigs from all samples within each host group. Individual binning results were refined using MAGScoT[23] to acquire high quality metagenome-assembled genomes (MAGs) for each sample. Clustering of MAGs to species-level genome bins (SGBs) was performed with dRep[84] in a multi-step approach to control for inflated SGBs due to low MAG quality. First, MAGs were dereplicated to 97% similarity within each host group, choosing the MAG with the highest score (calculated by MAGScoT based on completeness and contamination) as cluster representative. High and good quality representatives (score > =0.7) from all host groups together with representative sequences from the UHGG v2 were then clustered into 95% SGBs using dRep, again selecting the highest quality MAG as representatives. Medium quality (scores between 0.5 and 0.7) 97% representatives from previous clustering step were then compared to SGB representatives using fastANI[72], assigning MAGs with high similarity (>=95%) to the respective SGB. Medium-quality 97% representatives without hits to the high-quality SGB library were then clustered into 95% SGBs and added to the catalog in the case of at least two genomes in the cluster, discarding singleton clusters. The final catalog of SGB representatives was used to quantify contig abundances in all samples using Salmon in metagenome mode[85]. Taxonomic annotations were performed using the GTDBtk (v2.1) and GTDB release 207v2[86,87]. For SGBs without genus- and/or species-level assignments, the SGB ID was used as taxonomic label. GTDBtk marker gene alignments were used to generate a phylogenetic tree of all SGB representatives using the respective "infer" function of the GTDBtk. All data processing scripts are available online: https://github.com/mruehlemann/greatapes_mgx_scripts

### Pangenome catalog creation, annotation and analysis

All MAGs underwent calling of coding sequences using prodigal (v2.6.3)[88]. Protein sequences were clustered based on 95% similarity using MMseqs[89,90] and annotated using the emapper.py script of the eggNOG-mapper v2[91] annotation tool with the eggNOG 5.0 reference database[92]. MAG level functional profiles based on KEGG Ortholog annotations were collapsed into SGB-level pangenomes for each host genus (*Homo*, *Gorilla*, and *Pan*). In the case that no MAGs of an SGB were recovered from a given host genus, functional profiles were inferred from MAGs across the other host groups, accounting for host-specificities in the inferred accessory genomes/functions by considering a function to be present if it was present in all host-specific pangenomes of the respective SGB with MAGs recovered from the metagenomic data.

### Calculation of microbial clade and functional abundances

All downstream data processing and statistical calculations were performed in R v4.2[93] and using the tidyverse library[94]. Per-sample contig abundances for the SGB representatives from Salmon were used to estimate SGB abundances. Salmon output includes total mapped reads per contig and mapping reads adjusted for library size and total sequencing depth as transcripts per million (TPM), a measure from the transcriptomics field which can be directly transferred to metagenomic libraries. Individual contig coverages were calculated from the number of mapped reads and the effective lengths of the Salmon mapping output, considering contigs with >10% coverage as present. An SGB was considered present when at least 20% of its total length was in contigs marked as "present" and if at least 1,000 total reads and 250 TPM mapped to it. Final SGB abundances were calculated as TPM, calculated from the reads mapping to the SGBs present in the

respective sample, thus representing a normalized abundance across all samples. Combining SGB abundances with taxonomic assignments, domain- to species-level abundances were calculated as cumulative TPM abundances within the respective taxonomic bins. Rarefactions were calculated based on 5-fold repeated subsampling of contig level mapped reads at 100k, 250k, 500k, 1 M, 2.5 M, 5 M, and 10 M reads, followed by TPM calculations as described above. By rarefying reads and not TPM we realistically simulate sampling effects introduced by low coverage and low abundances of SGBs affecting especially samples with small library sizes. Community level functional profiles were calculated be multiplying TPM abundances of SGBs with the respective host-genus specific functional profiles (presence of KEGG orthologs [KOs]) of the SGBs and summarizing the per-SGB values into a sample-level abundance of functional annotations. Ultimately, functional abundances of individual KOs represent the cumulative TPM abundance of SGBs carrying the respective KO.

### Alpha and beta diversity

Faith's phylogenetic diversity (PD)[95] was used as measure of alpha diversity, calculated from the phylogenetic tree based on GTDBtk marker genes using the pd() function of the picante package for R[96]. Genus-level increase of PD from previously undetected SGBs was calculated from the differences of PDs with and without these SGBs annotated as the respective genus. Sample level PDs were calculated from the SGB presence/absence patterns. Beta diversity was assessed as unweighted and weighted UniFrac distances[97] using the UniFrac() function of the phyloseq package for R[98] and SGB abundances and the phylogenetic tree based on GTDBtk marker genes as input. Aitchison distance[99] was calculated from the CLR-transformed genus-level TPM abundances obtained from the clr() function from the compositions package for R[100] and adding a pseudocount of 1 to all abundances, setting all CLR-transformed abundances below zero to zero. Jaccard distances[101] were also calculated on genus-level presence/absence patterns using the vegdist() function from the vegan package for R[102]. Genus-level abundances were chosen for Aitchison and Jaccard distance, as SGBs are highly host-specific, thus would lead to high beta-diversities simply due to host exclusive SGBs, grouping at genus-level prevents from this and UniFrac distances use phylogenetic relations between SGBs. Beta-diversity on functional abundances were calculated from the Euclidian distances of the log-transformed KO abundances adding a pseudocount of 1 to avoid undefined values. Presence-absence values of KOs were treated in the same way as described above and using Jaccard distance to infer pairwise distances.

### Cross-host sharing analysis

Permutation-based analysis of excessive and reduced sharing of SGBs between host groups were based on the mean SGB abundances of the five rarefaction of 1 M mapped reads to account for differences in library depth impacting SGB richness and per-group sample sizes. For each host group, 100-fold sampling of five samples from this group were drawn and the SGBs found in the host were analyzed for their presence in five random samples of each of the other host groups, calculating the relative amount of shared SGBs as $Rel_{shared} = n_{SGB,shared} / n_{SGB,host}$. The mean of all 100 samplings was used as relative sharing coefficient for all host pairs in both directions. Excess and reduced sharing was analyzed by 1000-fold drawing of five random samples accounting for differences of host groups and the repetition of above calculations for relative sharing with all host groups. *P*-values were calculated from the proportions of random samplings exceeding/falling below the true sharing coefficients.

### Phylosymbiosis analysis

Phylosymbiosis was assessed using five measures for community level diversity, unweighted and weighted UniFrac, genus-level Aitchison, and Jaccard distances, as well as KEGG ortholog (KO) abundance based

Aitchison distance, and following the approach described in Brooks et al. (2016). Briefly, host group differences were used to infer microbiome dendrograms by UPGMA clustering. Branch support was calculated from 1000-fold jackknife sampling. Robinson-Foulds distances between microbiome trees and host phylogeny were calculated using the RF.dist() function from the phangorn package for R[103]. Significance of phylosymbiosis was assessed using the host phylogeny and 100,000 random trees as comparison for the microbiome trees. Tanglegrams were created with the ggtree and cowplot packages for R[104].

### Assessment of between-group abundance differences

Taxonomic abundance differences between Humans and NHAs, as well as between humans living outside and within industrialized systems were based on CLR-transformed abundances to account for the compositionality of microbiome data[105]. Included in the calculations were all genera with a prevalence >20% and relative abundance (before CLR-transformation) of >0.1% in at least of the host groups and all KO abundances with a prevalence >20% and CLR-transformed abundance of >1 in at least one of the host groups.

KO abundances were filtered accordingly and subsequently (log+1)-transformed to achieve a less skewed distribution. Log-transformation was chosen, as CLR-transformation assumes compositionality of the data, which is – unlike for taxonomic abundances - not fulfilled for functional abundances. Abundance differences were assessed in a linear regression in R[93] using abundances as dependent variable and human/NHA and European/African dichotomies as explanatory variables in a single model for each taxon (or function), assessing associations with all groups at once, the model was defined as lm(abundance ~ Human + European). $P$-values were calculated from the t-values of the resulting models using the summary.lm() function. Log-fold differences were calculated using group mean abundances and a pseudocount of 0.01. $P$-values were adjusted for multiple testing using Bonferroni correction. Features with significant ($Q < 0.05$) positive association with NHAs were grouped as "NHA associated". Features associating with geographic differences (Europe/Africa) were grouped into the respective group they were positively associated with. Remaining genera with significant differences between humans and NHAs, but not with a particular subgroup were grouped as "human associated". Genera without abundance differences in any of these comparisons were grouped as "unchanged" or "other".

### Functional pangenome differences between groups

Pangenome catalogs of human-associated SGBs were compared within microbial families between SGBs in taxonomic groups found enriched in European communities compared to other human-associated taxa, independent of a strong association with geography. KEGG Ontology (KO) term annotations were used as functional groups and their prevalence differences between groups were assessed using Fisher's exact test. Per-family effect sizes (Z-Scores) of KOs were calculated from $P_{Fisher}$-values and the direction of the effects which were assessed using the log2 of the ratio of prevalences in the two groups and a pseudo count of 0.01. The sum of the Z-Scores were added and divided by the square-root of the total number of families the respective KOs were found in to obtain a $Z_{Meta}$ for each KO term, used to calculated $P_{Meta}$. $P_{Meta}$-values were adjusted for multiple testing using Holm-correction. KO terms with $Q < 0.05$ and present in at least two of the microbial families in the analysis were considered as functions with differential prevalence. A similar approach was employed to assess functional differences between NHP- and Human-associated SGBs, however in this case, SGB pangenome differences were compared on genus level and the meta-analysis was performed combining signals from all genera, and specifically across the genera within particular phyla.

### Tree reconciliation analysis

Proteins from the representative SGBs of the genus *Prevotella* annotated with the annotation "cydA" (cytochrome *bd* oxidase subunit 1) as "Preferred name" in the emapper/eggNOG annotation were extracted from the unclustered protein sequence catalog. The same procedure was followed for *Paraprevotella clara*, which was included as an outgroup. Incomplete cydA sequences were removed using a length threshold of 200. Protein sequences were aligned using Clustal Omega[106]. The alignment was used to reconstruct the phylogenetic tree using IQTREE2[107] and a automatic model selection, which resulted in an LG + F + R8 model to be chosen as best-fit model according to the Bayesian information criterion (BIC). Branch support values were calculated using UFBoot[108] and performing SH-aLRT test[109]. Alignments of GTDBtk marker protein sequences for *Prevotella* SGBs and *Paraprevotella clara* were used to reconstruct a genome-level species phylogeny in the same respective way as described above for the cydA sequences (BIC best-fit: LG + I + I + R5). Low confident branches (<30% bootstrap support) in the cydA phylogeny were resolved together with the species tree using the OptResolutions supplementary program of the RANGER-DTL 2.0 software[110] resulting in 495 equally probable trees with optimized duplication-transfer-loss costs using default values (duplication: 2, loss: 1, transfer: 3). A randomly chosen output tree was using in the reconciliation analysis with the species tree in RANGER-DTL 2.0 using default values and 1000 random starting seeds in parallel[111] to assess robustness. Resulting sampling outcomes were summarized using the AggregateRanger tool of the RANGER-DTL 2.0 software package. For a global comparison of loss and transfer events, all *Prevotella* genes were selected that had (1) a 'PreferredName' annotation, (2) were found in at least 20% of *Prevotella* SGBs, and (3) occurred only as single copy per genome, resulting in 751 genes to which the workflow described above was applied analogously. As loss and transfer events are influenced by gene prevalence, mean loss and transfer frequencies were standardized by dividing them by the total number of SGBs they were found in.

### Co-phylogeny analysis

Host phylogenetic trees were obtained from the 10kTrees website ([112]; https://10ktrees.nunn-lab.org/). To assure high-quality microbial phylogenies for the co-phylogeny analysis, family-level maximum-likelihood trees were reconstructed from the GTDBtk marker gene alignments with the IQTREE2 software[107] and a WAG model including a random SGB outside the respective families as outgroups. Family level trees were rooted and for each SGB traced from tip to root to identify for each SGB the smallest subtree which covered 4, 5, 6, and 7 host groups. Combining information from all SGBs, the overall set of smallest trees to be included in the co-phylogeny analysis were identified, discarding subtrees for which the inclusion criterion was fulfilled already for a smaller tree starting from a different tip. In addition, subtrees spanning more than a single genus were excluded from the analysis, as divergence times of microbial genera predate divergence of the included hosts[71]. For all subgroups included in the analysis, maximum-likelihood distances and trees using a WAG model in IQTREE2 were inferred from the marker gene alignment of all MAGs assigned to the SGBs in the respective subgroups. Co-phylogeny of the subgroup was assessed by randomly selecting one MAG per host, calculating congruence with the host tree by Robinson-Foulds metric and by Mantel-test[42]. Tiplabels were permutated 999-fold and $P$-values calculated. This process starting from the random selection of one MAG per host was repeated 999 times to obtain final $P$-values. Family-level co-phylogeny ratios were calculated based on the ratio of SGBs within subtrees with co-phylogeny signal and total SGBs in the respective family that were included in the analysis. Enrichment of co-phylogeny for each microbial family was calculated by using Fisher's exact test on the SGBs in the analysis dividing them into four groups

based on family membership and being in a subtree with co-phylogeny signal. All *P*-values were adjusted using FDR correction.

## Correlation of microbial phenotypes with cophylogeny signals

SGB representative genome sequences were analyzed using the Traitar tool[113] to infer up to 67 microbial traits. A total of 43 inferred traits present in more than 50 and less than $1017 \cdot 50 = 967$ of the 1017 SGBs included in the cophylogeny analysis were analyzed for their association with cophylogeny signals. Using the R package lme4qtl[114], for each of the traits a mixed logistic regression model was fitted across for the 1017 SGBs, using signal of cophylogeny (binary trait) as dependent variable and the trait as bineray fixed effect explanatory variable, accounting for phylogenetic and taxonomic relatedness between SGBs by including a relatedness matrix and phylum-level categories as random effects in the model and using a binomal function with probit as link. The relationship matrix was calculated by using the cophenetic distance matrix from the SGB phylogeny, scaled to values between 0 and 1 by dividing by its largest distance. Accordingly, genome size in megabases and gene counts derived from the number of genes in the prodigal output were included as fixed-effect continuous traits. Effect sizes and *P*-values of the individual models were taken from the summary() function in R. *P*-values were adjusted for multiple testing by Bonferroni correction. Exemplary phylum-level differences in D-xylose utilization and gene counts between SGBs with and without cophylogeny signals were calculated using non-parametric Fisher's exact and Wilcoxon rank sum test, respectively.

## Statistics & Reprodicibility

All statistic computing was performed in R v4.2[93] and using the tidyverse library[94]. The used statistical tests are given in the respective subsections. No statistical method was used to predetermine sample size. Samples were excluded from the analysis based on low mapping rate to the library of species-level microbial representatives (below 1,000,000 reads). *P*-values < 0.05 were considered statistically significant, either adjusted or unadjusted depending on the analysis. The experiments were not randomized. The Investigators were not blinded to allocation during experiments and outcome assessment.

## Reporting summary

Further information on research design is available in the Nature Portfolio Reporting Summary linked to this article.

## Data availability

All metagenomic sequencing data is available via the NCBI BioProject accession IDs PRJNA692042, PRJNA539933, and PRJNA491335. The collection of 7700 metagenome-assembled genomes has been deposited in the European Nucleotide Archive, Accession: PRJEB68160. Source data are provided with this paper.

## Code availability

All code to process sequencing files to generate the presented results and manuscript figures is available via https://github.com/mruehlemann/greatapes_mgx_scripts.

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

## Acknowledgements

We thank the IKMB Microbiome and NGS laboratories for excellent technical support. We thank Sébastien Calvignac-Spencer and Grit Schubert for their valuable contributions to the analysis and the

manuscript. This study is an outcome of the Deutsche Forschungsgemeinschaft (DFG) Collaborative Research Center 1182 "Origin and Function of Metaorganisms" (https://www.metaorganism-research.com, Project number: 261376515, Projects A2 and Z2; AF and JFB) and received infrastructure support from the DFG Excellence Cluster 2167 "Precision Medicine in Chronic Inflammation" (PMI; Project number: 390884018; AF). JFG was supported by the DFG grant "The ecology and evolution of primate phageomes" (GO 3443/1-1: Project number: 453352748) and the DFG Research Group "Sociality and Health in Primates" (FOR2136; CA 1108/3-1; Project number: 244372499). Research at Kokolopori was supported by the Max Planck Society and Harvard University. Research at Ozouga and Taï National Park was supported by the Max Planck Society. Research at Loango was supported by US Fish and Wildlife Service, Tusk Trust, Berggorilla Regenwald Direkthilfe, and the Max Planck Society. Research at Bwindi was supported by the Max Planck Society. The survey of the Bwindi mountain gorillas was funded by the International Gorilla Conservation Programme coalition members with supplemental funding from the Wildlife Conservation Society. We are extremely grateful to the organizations that facilitated work at these field sites for sample collection: Bwindi - The mountain gorilla survey was conducted by the Uganda Wildlife Authority, l'Institut Congolais pour la Conservation de la Nature, the Rwanda Development Board, the International Gorilla Conservation Programme, the Max Planck Institute for Evolutionary Anthropology, Conservation Through Public Health, the Mountain Gorilla Veterinary Project, the Institute for Tropical Forest Conservation, and The Dian Fossey Gorilla Fund; Kokolopori - Vie Sauvage, the Bonobo Conservation Initiative; Loango - Agence Nationale des Parcs Nationaux, the Centre National de la Recherche Scientifique et Technique of Gabon; Taï National Park - the Centre Suisse de Recherches Scientifiques en Côte d'Ivoire and the staff members of the Taï Chimpanzee Project. We thank Taylor Hermes for support in the writing of the manuscript.

## Author contributions

J.F.B., F.H.L., and A.F. designed the research project; C.A.-K., C.B., T.D., J.-J.M.-T., M.M.R., M.S., R.M.W., K.Z., A.F., and F.H.L. built and maintained research infrastructure, performed field research, and provided materials. M.R., C.B., J.F.G, M.G., S.W., M.P., J.F.B., F.H.L., and A.F. analyzed data. M.R., B.M.H., C.B., J.F.G., M.G., M.P., M.U., J.F.B., F.H.L., and A.F. wrote the manuscript. All authors read, edited, and approved the submitted version of the manuscript.

## Funding

## Competing interests

The authors declare no competing interests.

## Additional information

[1]Institute of Clinical Molecular Biology, Kiel University, Kiel, Germany. [2]Institute for Medical Microbiology and Hospital Epidemiology, Hannover Medical School, Hannover, Germany. [3]Applied Zoology and Nature Conservation, University of Greifswald, Greifswald, Germany. [4]Helmholtz Institute for One Health, Helmholtz-Centre for Infection Research (HZI), Greifswald, Germany. [5]Epidemiology of Highly Pathogenic Microorganisms, Robert Koch Institute, Berlin, Germany. [6]Viral Evolution, Robert Koch Institute, Berlin, Germany. [7]Evolutionary Genomics, Max Planck Institute for Evolutionary Biology, Plön, Germany. [8]Institute of Experimental Medicine, Kiel University, Kiel, Germany. [9]Nutriinformatics Research Group, Institute for Human Nutrition and Food Science, Kiel University, Kiel, Germany. [10]Training and Research Unit for in Medical Sciences, Alassane Ouattara University / University Teaching Hospital of Bouaké, Bouaké, Côte d'Ivoire. [11]Comparative BioCognition, Institute of Cognitive Science, University of Osnabrück, Osnabrück, Germany. [12]National Institute for Biomedical Research, National Laboratory of Public Health, Kinshasa, Democratic Republic of the Congo. [13]Department of Primate Behavior and Evolution, Max Planck Institute for Evolutionary Anthropology, Leipzig, Germany. [14]Department of Human Evolutionary Biology, Harvard University, Cambridge, MA, USA. [15]Max Planck Institute for Evolutionary Anthropology, Leipzig, Germany. [16]Institute of Cognitive Sciences, CNRS UMR5229 University Lyon 1, Bron Cedex, France. [17]Taï Chimpanzee Project, CSRS, Abidjan, Côte d'Ivoire. [18]Institute of Biology, University of Neuchatel, Neuchatel, Switzerland. [19]School of Psychology & Neuroscience, University of St Andrews, St Andrews, Scotland, UK. ✉e-mail: m.ruehlemann@ikmb.uni-kiel.de; a.franke@mucosa.com

