## [Peer Review File · Nature Communications]

REVIEWER EXPERTISE

Reviewer #1. Microbiome / evolutionary genetics.

Reviewer #2. Microbiome / Metagenomics.

Reviewer #3. Microbiome in NHPs.

REVIEWER COMMENTS

Reviewer #1 (Remarks to the Author):

Review of: Functional host-specific adaptation of the intestinal microbiome in Hominids

Summary

Ruhlemann et al., surveyed and reconstructed fecal microbiome genomes using shotgun metagenomic sequencing data from two species of gorillas, three species of chimpanzees, bonobos, and human populations from Africa and Europe. In this comparative metagenomics study, they find evidence of phyllosymbiosis supported by microbial taxonomy, e.g., through various measures of beta diversity, but not through functional adaptations, e.g., Histidine metabolism. They also find evidence suggesting loss of human microbial diversity in the European cohort relative to the African Human cohort, consistent with what others have found in the field.

Positives

This is an impressively large body of work, including sequencing as well as in-depth taxonomic and functional analysis of a large range of hominid-associated fecal communities in the guts of African great apes and two human populations. Results and Analysis convincingly show that the human microbiome may be a derived trait, i.e., the Human microbiome differs from that of other Great Apes. However, there remain some concerns as elaborated on below:

Major Critiques

1. Congruence of gene vs species tree:

The authors claim that tree incongruence between *Prevotella* species phylogenies and *cydA* gene phylogenies provides supporting evidence of multiple individual events of gene loss and multiple individual events of horizontal gene transfer. However, the evidence provided is not strong enough for the claims being made here. Mismatch between gene and species trees, such as shown in Figure 2C, is not surprising in and of itself. Rather, we might expect that for any given species and gene tree, there will always be some level of incongruence between the topologies. In other words, for any gene tree for even a moderately recombining bacterium, we expect divergence from the species tree (please see <https://elifesciences.org/articles/65366>).

To strengthen this point, it would be helpful to include some form of control for this phylogenetic analysis to show that the highlighted result in Figure 2c truly is surprising enough to warrant a claim of convergent evolution. On line 411, it is mentioned that there is a high rate of gene transfer and gene loss, but it is unclear to what this is being compared to. To more strongly emphasize why this specific cophylogeny is noteworthy, please include some sort of control. For example, comparison with other randomly selected gene vs. species trees would be helpful.

2. Setting a baseline expectation for differential abundance

Similarly, we also note that many of these findings examine differential abundances of bacterial species between various cohorts, e.g., between those more associated with Humans and those more associated with Great Apes. To more strongly emphasize the magnitude of differential abundance, consider also showing a species which is found in both cohorts to provide a baseline expectation by which species with a high degree of differential abundance can be compared to.

3. Regarding co-phylogeny of spore-forming bacteria:

The claim that spore forming bacteria show less co-phylogeny is certainly very exciting. In Nayfach et al. 2016 Genome Research, a variant of this hypothesis was explored in which spore forming bacteria showed less vertical transmission from mothers to infants (please consider citing this paper to tie into the broader literature). One concern is that it is possible that the number of sporulation genes present is confounded by genome size, which to our understanding, has not been controlled for. In other words, longer genome lengths could be correlated with more sporulation genes. Could the authors please implement some sort of genome-length control for this analysis?

4. Similarities between African human and great ape microbiomes:

There are observed similarities in the associations of species in the microbiomes of African Humans and the great apes populations, which are also sampled from Africa. Some work needs to be done to control for shared environmental features, such as climate or food source – these are likely to impact the microbiome and this source of potentially confounding signals is not mentioned.

5. Replication of European findings across datasets:

For claims concerning the European cohort, much of the correlation may be specific to the German population being used. If these patterns, such as histidine metabolic pathway abundance, were replicated across a variety of European populations as opposed to a singular German population, then the claims made in this paper would be significantly strengthened.

Minor Critiques

1. Data pipeline:

A major contribution of this paper is the generation of novel MAGs. However, it is confusing as to what data is being used where. Lines 126-128 describe that 7,506 MAGs are reconstructed from 203 fecal samples of humans and NHP. From lines 135-136, it sounds like the MAGs referenced in lines 126-128 are then clustered along with data from the UHGG database, which includes both isolates and MAGs. Our assumption here is that some combination of newly sequenced data (referenced in lines 126-128) is used along with UHGG data in this study; yet it is unclear which data is what and used where. Suggestion: Please provide a diagram of what data went into what analysis and overall workflow.

2. Linear regressions:

There was some confusion in interpreting the linear regressions in figure 1. Our confusion stems from a lack of clarity in the Figure caption and in the figure axis labels. Figure 1i) is described as showing t-values (which suggests that the effect sizes are t-distributed), Figure 1j) has t-values on the y-axis, and describes effect sizes (which suggests that the effect sizes are t-distributed). In Figure 1j) grey points are used to describe non-significant sites.

However; Figure 1e) which comes earlier, is framed similarly to Figure 1i) and has the same color-scheming as Figure 1i) and Figure 1j). Does this imply that Figure 1e) is also showing t-distributed effect sizes?

Additionally, effect sizes were computed using a log₂ ratio of prevalence and a pseudocount of 0.01. Why was a pseudocount of 0.01 used, as opposed to a smaller count like 0.00001 – was this pseudocount added for plotting convenience due to the log₂ transformation?

In line 859 it is mentioned that linear regression is used to assess differences in abundance. It appears that species is the explanatory variable and difference in abundance is the response variable. On linear regression, we would like to see further clarification on the method used and the reasoning for the choices listed above. As an example, in Figure 1e, is this treating NHP vs. Humans and AFR vs. EU as covariates? It is currently not immediately clear.

3. Interpretation of Figure 1f

In figure 1f, is the y-axis truncated? It is unclear if points are not being shown here or if they are plotted but are not easily identifiable. Additionally, we noticed that only the top tail of the box-whisker plots are visible and that individual data seem to follow the same color-scheme as established later in Figure 1g. We suspect there may be a signal of differential abundance in *P. copri* given that it has the highest magnitude effect size of regression for AFR vs. EU as shown in Figure 1e. We suggest plotting these data on a log scale to make it easier to observe differences between low and high abundance data.

4. Depletion of Human-associated (AFR) microbial species in Human (EUR) hosts

For figure 1g, our interpretation is that you are using the mean CLR abundance to show that microbial species associated with Human (AFR) are depleted in the Human (EUR) population. This depiction was not immediately clear on its purpose and interpretation. Is there a different way you can show this depletion in the European Human population? We suggest breaking this plot up into a larger separate figure or set of subfigures where the information is more clearly available to interpret; as is it is difficult to disentangle the meaningful information.

5. Color-coding of Figure 1i

For figure 1i, it seems somewhat ad-hoc to assign color coding based on the x-y axis. We recommend a more quantitative clustering method.

6. Sporulation Dispersal strategy citation:

In line 483 there is a citation to Groussin et al., 2020 and Moeller et al., 2016, but you might want to additionally cite Hildebrand 2021: Dispersal strategies shape persistence and evolution of human gut bacteria.

7. Description of African human rural population used in this study:

Further clarification is needed regarding the inclusion of the African human rural population. The population is described as being from “rural, remote villages” and some of the analysis seems to be

done with the assumption that the population might have evolutionary signatures from hunter-gatherer societies; however, this is never explicitly clarified. Resulting analysis which may view African microbiomes as an intermediate or proxy between the microbiomes of great apes and the European cohort are then difficult to evaluate, given this lack of clarity.

Reviewer #2 (Remarks to the Author):

The authors have generated a substantial collection of stool metagenome datasets from non-human primates and accompany human metagenomes from different sites. They've created a large assemblage of MAGs derived from these metagenomes, especially from NHPs. Subsequently, they performed comparative genomics, seeking to understand the functional adaptations that correspond to the divergence between humans and NHPs. The data produced and the ensuing conclusions hold potential relevance for the broader field.

The manuscript could significantly improve with a more concentrated and structured storyline, alongside curtailing its speculative content, which currently relies on restricted analyses, many of which are not sufficiently explained.

Specific comments:

1. Overall, this manuscript appears to be a technical detour that, in my opinion, delves into an excess of subjects. I doubt the value of including numerous minor conclusions or speculations that don't substantially enhance our overall comprehension of microbiome adaptation.
2. It appears that there's a disparity in the sample collection across different geographic locations. Such variation could potentially introduce a substantial bias in the metagenomic results. It's helpful for the authors to discuss these impacts.
3. The method section seems to not include the removal of NHP host reads.
4. Line 149: "novelty" needs a definition, and it's better to replace novelty with another term.
5. Beginning at line 224, the authors delve into an analysis of shared taxa across host groups, with methods described from line 828. It appears to me that this analysis is a downsampling simulation of

the collected data. The way in which this simulation results in conclusions about enrichment or depletion of shared taxa remains unclear to me.

6. The conclusions made in the sections "Widespread changes in fecal..." and "Success of taxa.." are predominantly rooted in textual analyses of the annotations assigned to the KOs. From my perspective, these homology-based annotations provide only an approximate estimation of gene or pathway functions. It's not uncommon for the actual function of genes to deviate from the annotation. I would recommend that the authors pare down the extensive discussion on specific gene annotations, perhaps focusing more on a select group of annotated functional units that display the most potent statistical indicators.

7. Line 383, what's the definition of convergent adaptation?

Reviewer #3 (Remarks to the Author):

This paper compares the gut microbiota of humans and non-human apes using shotgun metagenomics and associated MAGs. The paper represents a rich great-ape microbiome dataset and is technically strong. Overall, I think the data are nicely analyzed and presented. My main suggestion is that the authors rephrase some of the interpretations to soften them somewhat.

One of the main potential issues I see is with the term 'humanization' of the microbiome as it is used - in comparison to all other NHPs. Because the authors only compare to other apes, I think they need to be careful with their language and not refer to all NHPs. One of the papers they cite demonstrates that human microbiomes actually diverge from those of other apes and converge towards those of cercopithecines. Since there are only ape samples in this paper, it is possible that some of the taxa that humans are losing or gaining compared to other apes are contributing to the convergence with other NHPs. While I don't think the authors need to get into this level of detail, I do think it is important that they not suggest the data show humans are different from all other NHPs but rather that they are unique among apes. Also, we need to remember that apes have likely experienced divergent evolution from humans since the LCA, and it is possible that some of the differences are not losses and gains only by humans but also by other apes.

I have marked some other places where I think the language could be altered in my detailed comments below. I think the discussion of inter-country differences in human microbiomes is good in the discussion but is over-simplified earlier in the paper, particularly given the number of human populations included. I also think differences in preservation of the German and African samples

needs to be acknowledged since this could contribute to blooms of aerobic microbial taxa which seem to dominate the German samples - are we sure this is not a technical effect?

Line by line comments:

Line 45: Non-human apes - be more specific than NHPs.

Line 60 - I am not sure this is true. Some of the earliest microbiome population studies targeted low-income countries (Malawi, Venezuela, Tanzania, etc.)

Line 68 - It might be helpful to define the HDI briefly

Figure 1 - The panels are ordered in an odd way. There are also so many panels they are small and hard to see - consider reducing.

Line 315: This might be worth explaining a little more

Line 339: Couldn't this just be a signal of high sugar and high protein diets in Germany?

Line 517: This is true so I struggle with the comparison between HDI and LDI populations above. Since it is only three populations, one of which is HDI, this seems like a pretty big generalization. I would tone down the language earlier in the paper.

Line 538: Community composition? Assembly refers to development usually, and there are not data describing that in this paper.

Line 724: It seems like the samples from Germany were less well-preserved than those in Africa. This may have something to do with all the oxygen-tolerance patterns observed. This is worth noting.

Dear Reviewers,

we want to thank you for your elaborate feedback on our manuscript and the very helpful comments and suggestions. As all three reviewers voiced concerns about including only a single cohort from high HDI regions in the analysis and the potential pitfalls and lack of generalizability connected to this, we decided that these concerns could only be cleared up through the inclusion of an additional dataset. We thus included a randomly chosen subset of an additional cohort from Denmark (CITATION; n=24 samples included) in the data processing and analysis. We are convinced that the addition of this dataset strengthens the general message and confidence in the presented results and hope the reviewers agree.

In addition, we re-evaluated some of our previous approaches, especially regarding the treatment of abundances of functional annotations (KEGG Orthologs) and decided that the previously used CLR-transformation is not appropriate for functional data, as the assumption of compositionality is – in contrast to the taxonomic composition of the microbiota – not warranted for these data. KO abundances are calculated by the addition of SGB abundances of SGBs carrying a specific function (KO). Ultimately, we decided to re-analyse these parts not with CLR-transformed abundances, but rather using log-transformed abundances instead.

While the points mentioned above did not have major influence on the overall outcome of the results of the article, it did change some of the association patterns, leading to a - in parts - completely re-written and improved manuscript.

Below you find the point-by-point response to the reviewer comments. Changes made to the manuscript text as response to the reviewers' comments are shown in red.

Thank you again on behalf of all authors.

REVIEWER EXPERTISE

Reviewer #1. Microbiome / evolutionary genetics.

Reviewer #2. Microbiome / Metagenomics.

Reviewer #3. Microbiome in NHPs.

REVIEWER COMMENTS

Reviewer #1 (Remarks to the Author):

Review of: Functional host-specific adaptation of the intestinal microbiome in Hominids

Summary

Ruhlemann et al., surveyed and reconstructed fecal microbiome genomes using shotgun metagenomic sequencing data from two species of gorillas, three species of chimpanzees, bonobos, and human populations from Africa and Europe. In this comparative metagenomics study, they find evidence of phyllosymbiosis supported by microbial taxonomy, e.g., through various measures of beta diversity, but not through functional adaptations, e.g., Histidine metabolism. They also find evidence suggesting loss of human microbial diversity in the European cohort relative to the African Human cohort, consistent with what others have found in the field.

Positives

This is an impressively large body of work, including sequencing as well as in-depth taxonomic and functional analysis of a large range of hominid-associated fecal communities in the guts of African great apes and two human populations. Results and Analysis convincingly show that the human microbiome may be a derived trait, i.e., the Human microbiome differs from that of other Great Apes. However, there remain some concerns as elaborated on below:

Major Critiques

1. Congruence of gene vs species tree:

The authors claim that tree incongruence between *Prevotella* species phylogenies and *cydA* gene phylogenies provides supporting evidence of multiple individual events of gene loss and multiple individual events of horizontal gene transfer. However, the evidence provided is not strong enough for the claims being made here. Mismatch between gene and species trees, such as shown in Figure 2C, is not surprising in and of itself. Rather, we might expect that for any given species and gene tree, there will always be some level of incongruence between the topologies. In other words, for any gene tree for even a moderately recombining bacterium, we expect divergence from the species tree (please see <https://elifesciences.org/articles/65366>).

To strengthen this point, it would be helpful to include some form of control for this phylogenetic analysis to show that the highlighted result in Figure 2c truly is surprising enough to warrant a claim of convergent evolution. On line 411, it is mentioned that there is a high rate of gene transfer and gene loss, but it is unclear to what this is being compared to. To more strongly emphasize why this specific cophylogeny is noteworthy, please include some sort of control. For example, comparison with other randomly selected gene vs. species trees would be helpful.

R: Thank you for the remark. We agree that the wording in this part of the manuscript might not have been full appropriate to convey our intended message or claims. We did not intend to show an increased (or decreased) level of gene transfers or losses for *cydA* in *Prevotella* or any higher/lower levels of incongruences in a global comparison. Rather we aimed to investigate *cydA*'s evolutionary history across and within hominids. We now changed this paragraph to a more neutral, less comparative tone, emphasizing the focus on *cydA*:

“We performed a tree reconciliation using a duplication-transfer-loss (DTL; (Kundu and Bansal, 2018)) model between the *Prevotella* and *cydA* phylogenies, which revealed frequent events of gene transfer ($T = 41.552$) between - also distant - branches of the *Prevotella* phylogeny and subsequent losses ($L = 23.52$) in the NHA-associated clades. Interestingly, the two *cydA*-carrying SGBs found in NHAs are phylogenetically distant, however, their *cydA* genes are highly similar and most likely the result of a transfer from one to the other. a high rate of gene transfer ($= 32.227$) and loss ($= 35.769$) driving the evolution of the *cydA* gene family. The gene transfer events and mappings were robust across 1,000 reconciliations with different starting seeds, with 89.2% of all events and 67.6% of all mappings found with 100% consistency. These results suggest that the enrichment of cytochrome bd ubiquinol oxidases observed in humans compared to NHPAs are the result of multiple individual reoccurring events of gene loss and horizontal gene transfer between *Prevotella* clades within the hominid gut.”

2. Setting a baseline expectation for differential abundance

Similarly, we also note that many of these findings examine differential abundances of bacterial species between various cohorts, e.g., between those more associated with Humans and those more associated with Great Apes. To more strongly emphasize the magnitude of differential abundance, consider also showing a species which is found in both cohorts to provide a baseline expectation by which species with a high degree of differential abundance can be compared to.

R: thank you, we agree that this part of the analysis deserves more detail in the figure. Following also the suggestion of Reviewer #3, we split figure 1 into multiple figures for better readability. The mentioned panel is now part of the new Figure 2, which in panel (b) displays exemplary abundance distributions for each of the four found associations (non-human apes (NHAs) vs Humans; Humans from Africa vs Europe) and on top two examples for taxa without strong associations to any single of the groups (*Eubacterium R* and *Methanobrevibacter A*). In addition, the increased available space made it possible for use to also show more clearly the individual subgroups in the analysis. We hope that this addresses the voiced concerns.

3. Regarding co-phylogeny of spore-forming bacteria:

The claim that spore forming bacteria show less co-phylogeny is certainly very exciting. In Nayfach et al. 2016 Genome Research, a variant of this hypothesis was explored in which spore forming bacteria showed less vertical transmission from mothers to infants (please consider citing this paper to tie into the broader literature). One concern is that it is possible that the number of sporulation genes present is confounded by genome size, which to our understanding, has not been controlled for. In other words, longer genome lengths could be correlated with more sporulation genes. Could the authors please implement some sort of genome-length control for this analysis?

R: thank you for bringing the article of Nayfach and colleagues to our attention in this context. As mentioned above, the addition of the additional dataset and changes in analysis resulted in this analysis no longer resulting in a significant association with any sporulation-associated phenotypes. Instead we replaced the analysis with a more generalizable approach analyzing in-silico inferred traits plus genome size and gene counts in relation to cophylogeny signals. We believe that this approach presents a much more valid and less arbitrary investigation of cophylogeny-related microbial traits. Results can be found in the respective paragraph and the new figure 5.

4. Similarities between African human and great ape microbiomes:

There are observed similarities in the associations of species in the microbiomes of African Humans and the great apes populations, which are also sampled from Africa. Some work needs to be done to control for shared environmental features, such as climate or food source – these are likely to impact the microbiome and this source of potentially confounding signals is not mentioned.

R: We agree that environment can have profound impact on the gut microbiota of hominids, however, we also think that the observed patterns in the Figures 1 and S2, together with the analysis of phylosymbiosis clearly show that evolutionary history and the time since divergence from a shared ancestor are much larger than such environmental influences.

5. Replication of European findings across datasets:

For claims concerning the European cohort, much of the correlation may be specific to the German population being used. If these patterns, such as histidine metabolic pathway abundance, were replicated across a variety of European populations as opposed to a singular German population, then the claims made in this paper would be significantly strengthened.

R: As mentioned in the starting paragraph to the Point-by-point reply, we agree that this was not address appropriately in the previous version of the manuscript. We now included an additional Danish cohort in the analysis and observe the same pattern for histidine metabolism / degradation. In addition we added a focused analysis of the prevalence of the enzyme urocanate hydratase (EC:4.2.1.49; K01712) which is responsible for the interconversion between urocanate and imidazole propionate in the histidine degradation pathway, which confirms our claims and the potential impact on metabolic health. We added a respective part to the manuscript:

“The enzyme urocanate hydratase (EC:4.2.1.49; K01712) is responsible for the interconversion between urocanate and imidazole propionate in the histidine degradation pathway. We find this gene significantly enriched in three of the four bacterial families analyzed ($P < 0.05$, Figure 3d), with a clear trend visible also for Oscillospiraceae ($P = 0.052$).”

Minor Critiques

1. Data pipeline:

A major contribution of this paper is the generation of novel MAGs. However, it is confusing as to what data is being used where. Lines 126-128 describe that 7,506 MAGs are reconstructed from 203 fecal samples of humans and NHP. From lines 135-136, it sounds like the MAGs referenced in lines 126-128 are then clustered along with data from the UHGG database, which includes both isolates and MAGs. Our assumption here is that some combination of newly sequenced data (referenced in lines 126-128) is used along with UHGG data in this study; yet it is unclear which data is what and used where. Suggestion: Please provide a diagram of what data went into what analysis and overall workflow.

R: thank you for the remark and the possibility to clarify. Indeed, we reconstructed (now) 7,700 MAGs from all samples ($n=227$, now including the Danish dataset) and added two large sets of reference SGBs/species. We added this overview as a panel to supplementary figure S1

In addition, we clarify the manuscript text:

„To ensure a comprehensive reference for the analysis, the collection of MAGs was combined with two large collection of microbial reference species reconstructed from human fecal metagenomes (UHGGv2, $n = 4,744$ isolates and MAGs; Almeida et al. 2021), and non-human primate fecal metagenomes ($n = 1,295$ MAGs; (Manara et al., 2019)), resulting in a total of $n = 13,739$ genome sequences, which were subsequently clustered into 5,777 species-level genome bins (SGBs; 95% ANI) using stringent criteria (Supplementary Figure S1a and S1b, Supplementary Table 1).“

2. Linear regressions:

There was some confusion in interpreting the linear regressions in figure 1. Our confusion stems from a lack of clarity in the Figure caption and in the figure axis labels. Figure 1i) is described as showing t-values (which suggests that the effect sizes are t-distributed), Figure 1j) has t-values on the y-axis, and describes effect sizes (which suggests that the effect sizes are t-distributed). In Figure 1j) grey points are used to describe non-significant sites.

However; Figure 1e) which comes earlier, is framed similarly to Figure 1i) and has the same color-scheming as Figure 1i) and Figure 1j). Does this imply that Figure 1e) is also showing t-distributed effect sizes?

R: thank you for bringing this to our attention. It is true, that there were some color mixups in the previous version of figure 1 which very likely lead to confusion while reviewing the manuscript – we sincerely apologize for the confusion this error lead to. We now corrected all plots to follow a consistent color scheme throughout the manuscript.

It is correct, that the plots 1e, h and j (now 2a, 3a and 3b) show the t-values from the linear regression to assess abundance differences between groups which indeed assumes t-distribution. In the taxonomic model, abundances are CLR-transformed, for the functional analysis, they are (now) log transformed (as also mentioned in the beginning of the PbP and updated in the methods section). We chose to show t-values for the figures as they are used to calculate P-values, while beta values / effect sizes are always also dependent on the standard errors of the regression.

Additionally, effect sizes were computed using a log2 ratio of prevalence and a pseudocount of 0.01. Why was a pseudocount of 0.01 used, as opposed to a smaller count like 0.00001 – was this pseudocount added for plotting convenience due to the log2 transformation?

R: we apologize, this part of the analysis was not correctly phrased in the methods part. The log2 ratio of prevalences was solely used to infer effect directions, not effect sizes. Thus, it makes no difference whether a pseudocount of 0.01 or 0.00001 was used. Absolute effect sizes were assessed from the P-values obtained from the Fisher's exact test and subsequently multiplied with the obtained effect directions (1 or -1). We now corrected this in the methods section

“Per-family **effect sizes (Z-Scores)** of KOs were calculated from P_{Fisher} -values and the direction of the **effects which were assessed using the log2 of the ratio of prevalences in the two groups and a pseudo count of 0.01.**”

In line 859 it is mentioned that linear regression is used to assess differences in abundance. It appears that species is the explanatory variable and difference in abundance is the response variable. On linear regression, we would like to see further clarification on the method used and the reasoning for the choices listed above. As an example, in Figure 1e, is this treating NHP vs. Humans and AFR vs. EU as covariates? It is currently not immediately clear.

R: Thank you for bringing this to our attention, we agree that this was not sufficiently clear in the methods section. We now clarify in the methods section:

“Abundance differences were assessed in a linear **regression in R (R Core Team, 2022) using abundances as dependent variable and human/NHA and European/African dichotomies as explanatory variables in a single model for each taxon (or function), assessing associations with all groups at once, the model was defined as $\text{lm}(\text{abundance} \sim \text{Human} + \text{European})$. P-values were calculated from the t-values of the resulting models using the `summary.lm()` function.**”

We used CLR-transformed taxonomic abundances to account for the compositionality of the data as mentioned in the methods section (“Taxonomic abundance differences [...] were based on CLR-transformed abundances to account for the compositionality of microbiome data (Gloor et al., 2017)”) As mentioned in the starting statement of the Point-by-Point reply, we re-evaluated our model choice for the functional abundances, for which we now used (log+1)-transformation to reduce distribution skewness, as we no longer considered CLR-transformation for this data appropriate. We elaborate in the methods section:

“KO abundances were filtered accordingly and subsequently (log+1)-transformed to achieve a less skewed distribution. Log-transformation was chosen, as CLR-transformation assumes compositionality of the data, which is – unlike for taxonomic abundances - not fulfilled for functional abundances.”

3. Interpretation of Figure 1f

In figure 1f, is the y-axis truncated? It is unclear if points are not being shown here or if they are plotted but are not easily identifiable. Additionally, we noticed that only the top tail of the box-whisker plots are visible and that individual data seem to follow the same color-scheme as established later in Figure 1g. We suspect there may be a signal of differential abundance in *P. copri* given that it has the highest magnitude effect size of regression for AFR vs. EU as shown in Figure 1e. We suggest plotting these data on a log scale to make it easier to observe differences between low and high abundance data.

R: we agree, the plots shown in figure 1f (now 2b) were not sufficiently clear, also due the aforementioned issues with incorrect color coding, which should be solved and consistent now in the updated plots and the entire manuscript – we apologize for the confusion resulting from this error. We hope that the updated plots now much better show the per-group distributions. All datapoints are shown in the respective plots, no values outside of the plotting area are truncated. Regarding the genus *Prevotella*, we hope it is now much clearer, that the analysis indicated a rather stable abundance between (all) humans and NHPs in general, however a significantly higher abundance in Humans from Africa compared to Europeans. We decided to not use log-transformed axes for these plots, but rather have individually scaled y-axes for each taxon, as data are partially sparse for subgroups and would give distorted depictions of the distributions especially in boxplots.

4. Depletion of Human-associated (AFR) microbial species in Human (EUR) hosts

For figure 1g, our interpretation is that you are using the mean CLR abundance to show that microbial species associated with Human (AFR) are depleted in the Human (EUR) population. This depiction was not immediately clear on its purpose and interpretation. Is there a different way you can show this depletion in the European Human population? We suggest breaking this plot up into a larger separate figure or set of subfigures where the information is more clearly available to interpret; as is it is difficult to disentangle the meaningful information.

R: thank you for the remark regarding figure 1g (now figure 2c). We changed this figure to now show cumulative relative abundances of taxa based on their association with a subgroup, depicted for each of the subgroups: NHAs, Humans (AFR) and Humans (EUR). Due to splitting the previous figure 1 into three separate figure, we agree, that the readability of the individual figures has much improved.

5. Color-coding of Figure 1i

For figure 1i, it seems somewhat ad-hoc to assign color coding based on the x-y axis. We recommend a more quantitative clustering method.

R: we apologize for the confusing color coding in figure 1. As mentioned above, we fixed this error and now (as also intended before) follow a consistent color scheme throughout all figures. Regarding the clustering method, we follow a strictly quantitative grouping by significant effects and effect direction. The horizontal and vertical lines included in the plots are displaying the t-value significance thresholds after Bonferroni-correction for multiple testing, which are $|t\text{-value}| > 4.04$ for the taxonomic analysis and $|t\text{-value}| > 4.77$ for the KO analysis. We included this information now in the figure legends of the (new) Figure 2a and 3a.

“Horizontal and vertical lines shown depict the t-value threshold ($|t\text{-value}| > 4.04$) for statistical significance after Bonferroni-correction.”

6. Sporulation Dispersal strategy citation:

In line 483 there is a citation to Groussin et al., 2020 and Moeller et al., 2016, but you might want to additionally cite Hildebrand 2021: Dispersal strategies shape persistence and evolution of human gut bacteria.

R: Thank you, we agree that Hildebrand et al. 2021 confirms these patterns as well and should be cited in this context. We added this reference to the respective part in the manuscript.

7. Description of African human rural population used in this study:

Further clarification is needed regarding the inclusion of the African human rural population. The population is described as being from “rural, remote villages” and some of the analysis seems to be done with the assumption that the population might have evolutionary signatures from hunter-gatherer societies; however, this is never explicitly clarified. Resulting analysis which may view African microbiomes as an intermediate or proxy between the microbiomes of great apes and the European cohort are then difficult to evaluate, given this lack of clarity.

R: thank you. We agree that the included cohort needed additional description and elaborated in the introduction.

“Additionally, we sequenced human fecal samples from two African populations (Mossoun et al., 2017) from rural villages of the Taï region in Côte d’Ivoire (HDI2021 = 0.550; (UNDP (United Nations Development Programme), 2022)) and the Bandundu region near Salonga National Park, Democratic Republic of the Congo (HDI2021 = 0.479), along with samples from Germany (HDI2021 = 0.942), and included a published dataset from Denmark (HDI2021 = 0.948; (Hansen et al., 2018)) to incorporate varying degrees of HDI.”

And the methods section:

“Sampling of wild-living great apes and human populations in Africa were conducted at: Bwindi Impenetrable Forest National Park, Uganda (Gorilla beringei beringi, Pan troglodytes schweinfurthii); Kokolopori Bonobo Reserve and villages adjacent to Salonga-Sud National Park, Democratic Republic of the Congo (Pan paniscus, Human); Loango National Park, Gabon (Gorilla gorilla gorilla, Pan troglodytes troglodytes); Taï National Park and adjacent villages, Côte d’Ivoire (Pan troglodytes verus, Human).”

However, we did not assume the microbiomes from the African populations as being an intermediate between great apes and European. Our intention was to analyze pattern of change between NHAs and Humans with an additional angle of geography (with differences in lifestyle, diet, cultural practices, HDI, etc. nested in this setup). Our results demonstrate that the change in microbiome composition and function between NHAs and Humans are strong, independent of their geographic origin.

Reviewer #2 (Remarks to the Author):

The authors have generated a substantial collection of stool metagenome datasets from non-human primates and accompany human metagenomes from different sites. They’ve created a large assemblage of MAGs derived from these metagenomes, especially from NHPs. Subsequently, they performed comparative genomics, seeking to understand the functional adaptations that correspond to the divergence between humans and NHPs. The data produced and the ensuing conclusions hold potential relevance for the broader field.

The manuscript could significantly improve with a more concentrated and structured storyline, alongside curtailing its speculative content, which currently relies on restricted analyses, many of which are not sufficiently explained.

R: We apologize if there was a lack of clarity in some of the analyses. We re-wrote large parts of the main text for better readability and elaborated on previously missing methodological details in the methods section. We hope this accounts for the doubts addressed by the reviewer.

Specific comments:

1. Overall, this manuscript appears to be a technical detour that, in my opinion, delves into an excess of subjects. I doubt the value of including numerous minor conclusions or speculations that don't substantially enhance our overall comprehension of microbiome adaptation.

R: thank you for this remark. We re-wrote large parts of the main text to highlight specific findings we found particularly worthy of discussion and added detailed analyses, e.g. on the topic of histidine metabolism. We hope the re-worked manuscript solves the reviewers issues with the article

2. It appears that there's a disparity in the sample collection across different geographic locations. Such variation could potentially introduce a substantial bias in the metagenomic results. It's helpful for the authors to discuss these impacts.

R: Thank you for the comment. This statement is in agreement with the other reviewers. We included an additional dataset of n=24 samples from a Danish cohort to reduce false-positive associations from differences in sample preservations when basing analyses on only a single European cohort. We added a statement on the differences in sample preservation in the Discussion part of the manuscript.

„In addition and due to logistic constraints, preservation methods for fecal samples from the included hosts and host subgroups differ, making it impossible, to entirely disentangle technical artifacts from true biological effects

3. The method section seems to not include the removal of NHP host reads.

R: Decontamination of NHA metagenome were also performed using the same database as for humans, all with a lenient threshold of 95% identity, which should capture the majority of host contaminant reads. We clarified this in the methods section:

“Host reads were removed using bbmap.sh. A masked human reference database (“Introducing RemoveHuman,” 2014) and a lenient mapping threshold of 95% identity was used to account for a broader host range to also capture host contamination from the Pan and Gorilla host.”

Ultimately, even if host reads were not removed entirely, this would likely not impact abundance estimates, as they are based on the mapping to the non-redundant collection of SGBs, which would not be biased by any remaining host reads, as they would not map to the database. It would potentially lower the overall mapping rate, which was used as an exclusion criterion, thus – even if there are host reads remaining in the data – the influence on the results is expected to be negligible.

4. Line 149: “novelty” needs a definition, and it’s better to replace novelty with another term.

R: thank you, we agree that the term “novel” should not be used in this context. We replaced the term with “not previously covered by any of the two large reference sets”

5. Beginning at line 224, the authors delve into an analysis of shared taxa across host groups, with methods described from line 828. It appears to me that this analysis is a downsampling simulation of the collected data. The way in which this simulation results in conclusions about enrichment or depletion of shared taxa remains unclear to me.

R: The analysis of shared clades across different host groups was designed to account for differences in (a) sequencing depths and (b) group sizes by the utilization of a permutation-based framework. Both factors are prone to influence diversity estimates. The former was taken care of by using presence/absence estimates at a normalized sampling depth of 1 Million reads, the latter by randomly drawing five samples from each of the groups in the analysis. We added an introductory sentence to the respective paragraph:

“SGBs shared between host groups were analyzed in a permutation-based framework accounting for differences in sequencing depth and group sizes (see Methods for details).”

6. The conclusions made in the sections "Widespread changes in fecal..." and "Success of taxa.." are predominantly rooted in textual analyses of the annotations assigned to the KOs. From my perspective, these homology-based annotations provide only an approximate estimation of gene or pathway functions. It's not uncommon for the actual function of genes to deviate from the annotation. I would recommend that the authors pare down the extensive discussion on specific gene annotations, perhaps focusing more on a select group of annotated functional units that display the most potent statistical indicators.

R: thank you for pointing this out. We agree that individual annotations are likely biased by databases and might not fully reflect true functions. We (re-analysed and) re-wrote the respective paragraphs to focus on enrichment/depletion of higher-level KEGG annotations (pathways, module, etc.) for which our statistical approaches are more likely to pick up true positive signals through the need of multiple individual signals to be present. Abundance / prevalence differences of individual KOs are reported in the Supplementary Tables S6-9.

7. Line 383, what's the definition of convergent adaptation?

R: We added a brief explanation of “convergent adaptation” in the respective part of the manuscript:

“Shared gene gains or losses across multiple microbial clades can indicate a response to specific host intestinal environments, leading to functions being acquired (and selected for) independently multiple times. We performed a pangenome analysis of genera shared between humans and NHAs (n=36) to identify such patterns of convergent adaptation.”

Reviewer #3 (Remarks to the Author):

This paper compares the gut microbiota of humans and non-human apes using shotgun metagenomics and associated MAGs. The paper represents a rich great-ape microbiome dataset and is technically strong. Overall, I think the data are nicely analyzed and presented. My main suggestion is that the authors rephrase some of the interpretations to soften them somewhat.

One of the main potential issues I see is with the term ‘humanization’ of the microbiome as it is used - in comparison to all other NHPs. Because the authors only compare to other apes, I think they need to be careful with their language and not refer to all NHPs. One of the papers they cite demonstrates that human microbiomes actually diverge from those of other apes and converge towards those of cercopithecines. Since there are only ape samples in this paper, it is possible that some of the taxa that humans are losing or gaining compared to other apes are contributing to the convergence with other NHPs. While I don't think the authors need to get into this level of detail, I do think it is important that they not suggest the data show humans are different from all other NHPs but rather that they are unique among apes. Also, we need to remember that apes have likely experienced divergent evolution from humans since the LCA, and it is possible that some of the differences are not losses and gains only by humans but also by other apes.

R: We want to thank the reviewer for this highly important remark. We strongly agree, that the use of the term “non-human primates” in our manuscript (and its implications) is not accurate and replaced it with the correct, more accurate term non-human apes (NHA). We think this already solves many of the implied assumptions stated throughout the manuscript's text. Additionally, we added a brief paragraph to the discussion specifically addressing the limitations of our study with regard to higher-level ecological impacts and the convergence towards cercopithecines:

“Additional limitations stem from the focus on humans and African great apes. While the comparisons between these host-clades provide a framework for the in-depth investigation of (evolutionary) rather recent adaptations and between-host divergences, they potentially neglect that could be revealed by broader-scale investigations, such as the previously described convergence of the human gut microbiota towards that of cercopithecines (Amato et al., 2019). However, our analyses show the impact of the unique trajectories taken by the intestinal microbiomes of Pan and Homo since their last common ancestor.

We also strongly agree that it's important to keep a neutral language without a (too much emphasized) anthropocentric view on results. We now entirely removed the term "humanization" from the manuscript, replacing it with more appropriate formulations.

Lastly, we assured to only talk about "loss" or "gain" of taxa when a clear trajectory is visible, e.g. the loss of evolutionary conserved *Prevotella* diversity/SGBs in humans from Europe, or the expansion of *Bacteroides* in humans from Europe – a clade that is virtually absent across NHAs.

I have marked some other places where I think the language could be altered in my detailed comments below. I think the discussion of inter-country differences in human microbiomes is good in the discussion but is over-simplified earlier in the paper, particularly given the number of human populations included. I also think differences in preservation of the German and African samples needs to be acknowledged since this could contribute to blooms of aerobic microbial taxa which seem to dominate the German samples - are we sure this is not a technical effect?

R: Thank you for pointing this out. As also mentioned above, we included an additional dataset from Denmark to reduce the effect of technical artifacts driven by single individual cohort specificities. We also now mention this drawback in the discussion:

"Due to logistic constraints, preservation methods for fecal samples from the included hosts and host subgroups differ. While these are expected to influence microbiome composition, previous studies show that individual signature are retained independent of storage methods (Blekhman et al., 2016).."

Line by line comments:

Line 45: Non-human apes - be more specific than NHPs.

R: Thank you for bringing this up, we agree that Non-human apes (NHAs) is the appropriate term and changed accordingly throughout the manuscript

Line 60 - I am not sure this is true. Some of the earliest microbiome population studies targeted low-income countries (Malawi, Venezuela, Tanzania, etc.)

R: thank you, you are correct. We changed the sentence to "To date, the majority of these studies were conducted on sample collections from high-income countries, however growing efforts to include humans from diverse global populations are underway, [...]"

Line 68 - It might be helpful to define the HDI briefly

R: thank you, we agree and added a brief definition in the introduction at the first mention of HDI: "Human Development Index (HDI; a statistical composite index of indicators covering life expectancy, education, and income)"

Figure 1 - The panels are ordered in an odd way. There are also so many panels they are small and hard to see - consider reducing.

R: thank you, we split figure 1 into the new figure 1-3, allowing for better readability and more detail in the displayed results.

Line 315: This might be worth explaining a little more

R: thank you, we added a bit more context to that part. "The increased abundance of Vitamin B12-producing microorganisms in the feces of Europeans may be driven by higher dietary intake of meat and dairy products from ruminants, as these food groups are known to contain microorganisms with this metabolic capacity, unlike edible plants and mushrooms (Watanabe and Bito, 2018)"

Line 339: Couldn't this just be a signal of high sugar and high protein diets in Germany?

R: Thank you for pointing this out. We are convinced, that specifically the higher potential for oxidative breakdown of carbohydrates might be the key finding here. However, we do not know the exact reasons and mechanisms behind it and agree that this might (and likely will be) driven by the higher intake of

especially carbohydrates. we clarified in the text, that this might be a potential reason for the seen signal:

„These signals strongly suggest that the selection for taxa in the gut of humans living in Europe is connected to a diet rich in carbohydrates and potentially the adaptation to transient microaerobic conditions in the gut environment, using oxidative phosphorylation as a mean to release energy from nutrients, which is more efficient than strictly anaerobic fermentation (Jurtshuk, 1996).“

Line 517: This is true so I struggle with the comparison between HDI and LDI populations above. Since it is only three populations, one of which is HDI, this seems like a pretty big generalization. I would tone down the language earlier in the paper.

R: Thank you for the remark. As mentioned above, we now added a second high HDI population from Denmark. In addition, we toned down the language in the manuscript regarding high/low HDI, focusing more on geographic differences.

Line 538: Community composition? Assembly refers to development usually, and there are not data describing that in this paper.

R: We agree and changed the text accordingly

Line 724: It seems like the samples from Germany were less well-preserved than those in Africa. This may have something to do with all the oxygen-tolerance patterns observed. This is worth noting.

R: Thank you for pointing this out. As mentioned above, we included an additional dataset from Denmark to decrease the influence of technical aspects on the presented comparative analyses. Additionally, we added a statement to the discussion:

„ Due to logistic constraints, preservation methods for fecal samples from the included hosts and host subgroups differ. While these are expected to influence microbiome composition, previous studies show that individual signature are retained independent of storage methods (Blekhman et al., 2016).“

REVIEWER COMMENTS

Reviewer #1 (Remarks to the Author):

We thank the authors for carefully considering our feedback. We overall were satisfied with the revisions made, except in response to our first point regarding the congruence of the species vs gene tree. We understand that the wording has been changed, however, this story still is quite prominent in the paper, warranting a comparison with randomly selected genes so that the patterns observed at the *cydA* locus can be contextualized. Additionally we see that now this figure is in Figure 4C with some modifications, including inclusion of arrows. We could not ascertain from the results or methods how the directionality and thickness of these arrows were inferred. Apologies if we missed this.

Reviewer #2 (Remarks to the Author):

The authors have done a good job addressing my questions and comments.

Reviewer #3 (Remarks to the Author):

My comments have been sufficiently addressed.

We thank all reviewers and the editor again for their valuable feedback leading to an improved manuscript. We are happy that our previous reply addressed most of the raised points and hope that the new version now can now also address the remaining open remarks.

In addition, as requested by the editor, we now re-edited the ethics and inclusion statement in the methods section and additionally were finally able to upload the complete catalog of metagenome-assembled genomes to the ENA servers where it will be available via the accession PRJEB68160. We included this information in the data availability statement.

Reviewer #1 (Remarks to the Author):

We thank the authors for carefully considering our feedback. We overall were satisfied with the revisions made, except in response to our first point regarding the congruence of the species vs gene tree. We understand that the wording has been changed, however, this story still is quite prominent in the paper, warranting a comparison with randomly selected genes so that the patterns observed at the *cydA* locus can be contextualized. Additionally we see that now this figure is in Figure 4C with some modifications, including inclusion of arrows. We could not ascertain from the results or methods how the directionality and thickness of these arrows were inferred. Apologies if we missed this.

R: We thank the reviewers again for their previous feedback and are happy we could largely address them. For the remaining open points regarding the contextualization of *cydA* loss and transfer frequencies, we extended the analysis slightly and modified it to include all single-copy genes occurring in at least 20% of *Prevotella* SGBs (n=752 incl. *cydA*). We added the details to the methods section and updated the main text, which now reads:

“We performed a tree reconciliation using a duplication-transfer-loss (DTL; (Kundu and Bansal, 2018)) model between the *Prevotella* and *cydA* phylogenies, which revealed frequent events of gene transfer ($\bar{T} = 44.6$) between branches (including distant ones) of the *Prevotella* phylogeny and subsequent losses ($\bar{L} = 23.2$) in the NHA-associated clades. **Standardized by gene prevalence, these values are in the 71st and 61st percentile for transfers and losses of single-copy genes found in the *Prevotella* genus, respectively (Suppl. Table S13).**”

We agree with the reviewers, that this is a valuable addition to the analysis, providing important context to the reconciliation results.

Regarding updated figure 4C, we thank the reviewers for bringing this point to our attention and giving us the opportunity to update the figure legend. Red arrows represent gene transfer events from a donor to a recipient node found in at least 50% of the 1000 reconciliations with random starting seeds and are weighted by the frequency of occurrence. We rewrote the figure legend to clarify this point, it now reads:

“[...] (c) Results of the tree reconciliation analysis for the *cydA* gene in *Prevotella* SGBs found in humans (blue) and NHAs (Pan: orange; Gorilla: green) demonstrate

a history of frequent transfer events across 1,000 reconciliations with random seeds. Filled and empty shapes represent cydA-positive and -negative SGBs, respectively. Red arrows depict gene transfer events found in at least 50% of reconciliations and are weighted by frequency. Red triangles mark nodes that were identified as gene transfer recipients with > 50% frequency independent of the donor node. Black circles mark speciation events with > 50% frequency. The ten highest abundant Prevotella species with established names are shown for orientation. The Prevotella tree was rooted using Paraprevotella clara as the outgroup (not shown).”

Reviewer #2 (Remarks to the Author):

The authors have done a good job addressing my questions and comments.

Reviewer #3 (Remarks to the Author):

My comments have been sufficiently addressed.

REVIEWERS' COMMENTS

Reviewer #1 (Remarks to the Author):

Thank you for attempting to address our comments, especially the tree reconciliation issue.

We thank all reviewers and the editor again for their valuable feedback to our manuscript.

REVIEWERS' COMMENTS

Reviewer #1 (Remarks to the Author):

Thank you for attempting to address our comments, especially the tree reconciliation issue.

A: thank you, we are happy we could address all remaining open comments